# Inferring residue level hydrogen deuterium exchange with ReX
Oliver M. Crook [1,2] ✉, Nathan Gittens[3], Chun-wa Chung [3] & Charlotte M. Deane [4]

Hydrogen-Deuterium Exchange Mass-Spectrometry (HDX-MS) has emerged as a powerful technique to explore the conformational dynamics of proteins and protein complexes in solution. The bottom-up approach to MS uses peptides to represent an average of residues, leading to reduced resolution of deuterium exchange and complicates the interpretation of the data. Here, we introduce ReX, a method to infer residue-level uptake patterns leveraging the overlap in peptides, the temporal component of the data and the correlation along the sequence dimension. This approach infers statistical significance for individual residues by treating HDX-MS as a multiple change-point problem. By fitting our model in a Bayesian non-parametric framework, we perform parameter number inference, differential HDX confidence assessments, and uncertainty estimation for temporal kinetics. Benchmarking against existing methods using a three-way proteolytic digestion experiment shows our method's superior performance at predicting unseen HDX data. Moreover, it aligns HDX-MS with the reporting standards of other structural methods by providing global and local resolution metrics. Using ReX, we analyze the differential flexibility of BRD4's two Bromodomains in the presence of I-BET151 and quantify the conformational variations induced by a panel of seventeen small molecules on LXRa. Our analysis reveals distinct residue-level HDX signatures for ligands with varied functional outcomes, highlighting the potential of this characterisation to inform mode of action analysis.

Understanding the conformational dynamics of proteins and the conformational changes induced by their interactions is key to determining their function. One method to explore such dynamics is hydrogen-deuterium exchange mass-spectrometry (HDX-MS), which tracks the rate of exchange of amide hydrogens with deuterium in solution[1–6]. This rate is impacted by various factors such as secondary (and higher order) structures, as well as solvent accessibility[7–10]. It is fundamentally based on the concept of local folding and unfolding, as suggested in the Linderstrøm-Lang theory[11]. The numerous applications of HDX-MS can include: the characterisation of binding interfaces of antibody-antigen complexes by mapping of their epitopes and paratopes[12]; examining structural stabilisation by small molecules and their corresponding allosteric effects[13]; and the effect of amino-acid mutations on transporters[10] and applications to vaccine design[14].

The majority of HDX-MS studies involve a bottom-up approach to mass-spectrometry[5]. In this widely employed strategy, proteins undergo on-line proteolytic digestion via a non-specific protease - often pepsin or nepenthesin-II - immobilized to a column, usually yielding good sequence coverage by reporting deuterium uptake at the peptide fragment level.

Although the selection of protease can be tailored to specific applications, inherent limitations emerge due to the resolution loss at the peptide level, obstructing a nuanced interpretation of the molecular phenomena under scrutiny. Hence, bottom-up HDX-MS leads to distinct scenarios that remain unaddressed by current statistical methodology. First, combining information from overlapping peptides (often referred to as redundancy) and replicates lacks a clear approach. Second, instances where overlapping peptides exhibit a mixture of protection and de-protection effects require a means of resolution, which is currently addressed by somewhat arbitrary approaches such as data omission[15]. Third, differences in the length of the overlapping peptides introduces interpretation issues, especially when contrasting modest, consistent effects on longer peptides against more variable effects on shorter ones. We refer to this phenomenon as the length-bias issue. Finally, these challenges need to be considered while also accounting for the temporal nature of HDX data and correlations along the sequence dimension.

Projecting peptide-level data to residue-level data is an apparent solution to the previously outlined challenges, and several strategies have been proposed in this direction[16–31]. These proposed approaches can be

[1]Department of Chemistry, Dorothy Crowfoot Hodgkin Building, University of Oxford, Oxford, UK. [2]Kavli Institute for Nanoscience Discovery, University of Oxford, Oxford, UK. [3]Structural and Biophysical Sciences, GlaxoSmithKline R&D, Stevenage, UK. [4]Department of Statistics, University of Oxford, Oxford, UK. ✉e-mail: oliver.crook@chem.ox.ac.uk

divided into non-optimisation approaches (e.g. ref. 16), such as taking a running average along the sequence dimension[31], and optimisation-based approaches where a model is developed and then fit to the data by minimising the discrepancy between the model and data (e.g. refs. 26,28–30). Such models can incorporate physically relevant constraints, such as prohibiting negative deuterium uptake. While a residue-level model, in principle, could overcome the aforementioned issues, it introduces its own set of challenges. The main obstacle is the underdetermination of an appropriate model, implying a scenario where model parameters outnumber data points. Further, HDX suffers from a more extreme form of underdetermination that cannot be mitigated by acquiring additional data. This arises due to the potential countless combinations of deuterium uptake for two adjacent residues, each with unit redundancy, as long as the sum of the uptake is consistent with the observed data. All of these drawbacks become exacerbated when one wishes to study multiple protein states and perform differential HDX-MS experiments. Here, it becomes paramount to associate a measure of statistical confidence to the results to avoid false positives. Currently, there is no appropriate statistical framework that attempts to tackle these issue.

In response to these outstanding methodological challenges, we developed ReX—a statistical method complemented by an associated software package, designed to achieve residue-resolved HDX. ReX combines a likelihood model, which models the deuterium uptake per residue, with a prior change-point model that permits correlations or jumps between the parameters of adjacent residues. ReX is a non-parametric Bayesian model (see ref. 32 for examples), meaning it does not fix the number of parameters, allowing model complexity to expand with the quality and volume of data. The Bayesian framework allows us to associate probability distributions to pertinent quantities (such as model parameters) that are updated in the light of data[32]. This updated distribution, termed the posterior distribution, allows us to quantify the uncertainty in our model inferences.

Despite a number of computational approaches available to obtain residue-level HDX from overlapping peptide-level HDX data, a comprehensive benchmarking of such approaches is absent from the literature. Directly conducting residue-level HDX via nuclear magnetic resonance (NMR) or electron transfer dissociation (ETD) has its own set of drawbacks[33,34]. In particular, NMR has low sensitivity while ETD has poor efficiency and typically needs sample-specific optimisation. Consequently, comparisons leveraging different experimental platforms are likely to only report on qualitative differences. Here, we propose a simple and extensible benchmark using a triple digestion of Cytochrome C. In this approach, we test the ability of different methods to reconstruct the HDX-MS data from the unseen digestions and quantify the error. In this experiment, we expect the same underlying HDX results, but by using different peptide digestions to generate independent datasets allows for a critical examination of whether diverse methods yield consistent residue-resolved HDX-MS profiles.

In this manuscript, we first detail the ReX model specification and posterior inference. We proceed to explain and perform a benchmarking analysis, highlighting ReX's superior performance but also providing insights into the recommendations and potential pitfalls associated with alternative methods. Subsequently, we illustrate how ReX can evaluate the resolution and quality of HDX data, as well as evaluate the consistency of structure prediction methods such as AlphaFold[35]. We further showcase ReX's capabilities through a variety of examples and extensive simulations. A comparative analysis of the stabilisation of the two bromodomains of BRD4 using I-BET151 demonstrates that the BC loop of the second bromodomain is more flexible than that of the first bromodomain. We then perform conformational signature analysis (CSA) on LXRα with seventeen small molecules at the residue-level. Here, we demonstrate the additional insights that can be drawn by quantifying uncertainty in the conformational landscape induced by the small molecules. The associated R-package (https://github.com/ococrook/RexMS) enables single-protein analysis (https://ococrook.github.io/RexMS/articles/ReX.html), differential analysis (https://ococrook.github.io/RexMS/articles/DifferentialRexMS.html) and conformational signature analysis (https://ococrook.github.io/RexMS/articles/

ConformationalSignatureAnalysis.html). The package comes with a variety of visualisations including PDB structure visualisation. We also provide Shiny Application for interactive data visualisation and as a template so other may construct their own. To interact with the BRD4 and LXRα results, visit the following webpages: https://olivercrook.shinyapps.io/BRD4-ReX/ and https://olivercrook.shinyapps.io/ConformationalSignatureAnalysis/.

## Results

### The ReX method and posterior inference

ReX is designed as a versatile statistical method aiming to facilitate the inference of residue-level HDX-MS across a range of experimental designs. It accommodates multiple protein states, including mutations, small molecules, complexes and more (see Fig. 1). While replicates and many time-points are not a strict requirement of the input data, we advocate for them. Incorporating additional replicates and time-points, especially those spanning three or four orders of magnitude, not only helps ReX infer the model parameters but also enhances confidence and quality in the outcomes. To ensure consistency between peptides, ReX incorporates information from fully deuterated controls.

Central to ReX is a Bayesian change-point model, designed to perform sub-localisation of HDX uptake. We posit the existence of a latent (unobserved), residue-level process, analogous to assumptions entailed in a Hidden Markov Model[36]. For a given residue, there is a temporal uptake model parametrised as mixture between a stretched exponential (Weibull-type) model[37] and a standard exponential (logistic model). The mixture proportions are learned during the inference process, enabling the model to better adapt without necessitating pre-specification. Given that this model usually possesses an excessive number of parameters, regularization of the parameter space is performed by defining a prior model[32,38]. The complexity of the inferred model is guided by the data, and thus, we non-parametrically specify the prior model as a change-point process[39,40]. Here, parameters between segments of residues are similar or necessitate a jump. This implies that the number of change points, their locations, and associated kinetic parameters must be inferred (see Fig. 1). While these quantities could be pre-specified, they are typically unknown, and hence, we include these quantities in the inference process. A prior Poisson process is employed to model the unknown (random) locations of these change points along the data's residue axis[41].

Inference in Bayesian models typically employs Markov-chain Monte-Carlo (MCMC) to draw samples from the posterior distribution of parameters, by using a Markov-Chain[42]. However, a challenge arises when adding or removing a change point which increases or decreases the dimension (i.e. the parameter count) of the model. Therefore, we developed an MCMC method that allows the Markov-chain to explore while increasing and decreasing the model's dimension. The Reversible Jump MCMC (RJMCMC), sometimes referred to as the trans-dimensional MCMC algorithm, facilitates such dimension switching[39]. This algorithm handles scenarios where the number of unknown entities is itself unknown. The intricacies are primarily in determining the transition probability when altering dimensions (see 'Methods' for more details).

Bridging the unobserved model to the data remains challenging. The observation model takes the residue-level model and adds up the residue values from the exchangeable residues covered by a peptide to derive peptide-level inferences, which can be compared with the data. The residuals are posited to follow a Laplace distribution with the scale (variance) parameter multiplied by the length of the peptide. This approach down-weights the contribution of longer peptides, or more precisely, permits them to be more variable. The RJMCMC algorithm samples from the posterior distribution of this model which we summarise using the posterior mean.

### An extensible benchmark for residue-level HDX-MS

In order to assess ReX, in comparison with other approaches, we devised a versatile and extensible experimental benchmark, designed to be straightforwardly implemented and extended by all HDX-MS practitioners. Taking

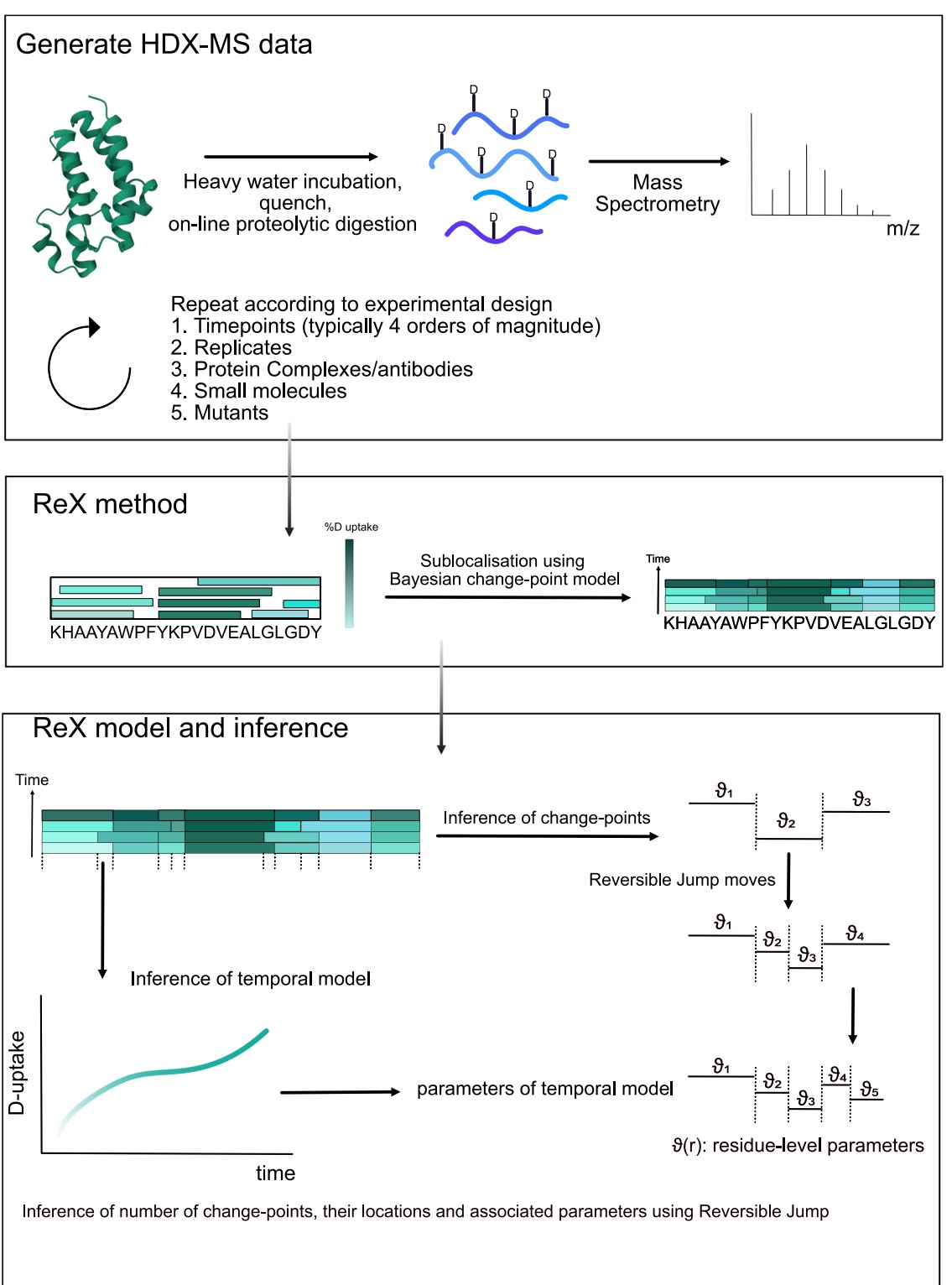

**Fig. 1 | An Overview of the ReX method, model and inference.** ReX is applicable to general HDX-MS experiments performed using the bottom-up approach. Typical variations in workflow are allowed including changes to temperature, pH, buffer concentration. Experimental designs with at least three labelling time points and replicate measurements enhances the quality of ReX's outputs. The method can be used for any protein states such as protein complexes, antibodies, small molecules and mutants as examples. The goal of ReX is to take the peptide map generated by the experiment with deuterium reported on peptide fragments and infer (sub-localise) the deuterium exchange measurements at the level of residues. ReX uses a Bayesian change-point model on the parameters of the temporal kinetics. This non-parametric model is fitted using reversible jump Markov-chain Monte-Carlo.

cytochrome-C, HDX was conducted at four labelling time points (15, 150, 1500, 15,000 s), initially undergoing digestion with nepenthesin II (Affipro) for three replicates. Subsequent digestions employing pepsin and aspergil-lopepsin (Type XIII) were performed in separate experiments. Each digest generates different peptide maps and we refer to this as a differential digestion experiment (see Fig. 2).

To computationally profile diverse approaches, residue-level uptake was initially inferred for each model under consideration, using data

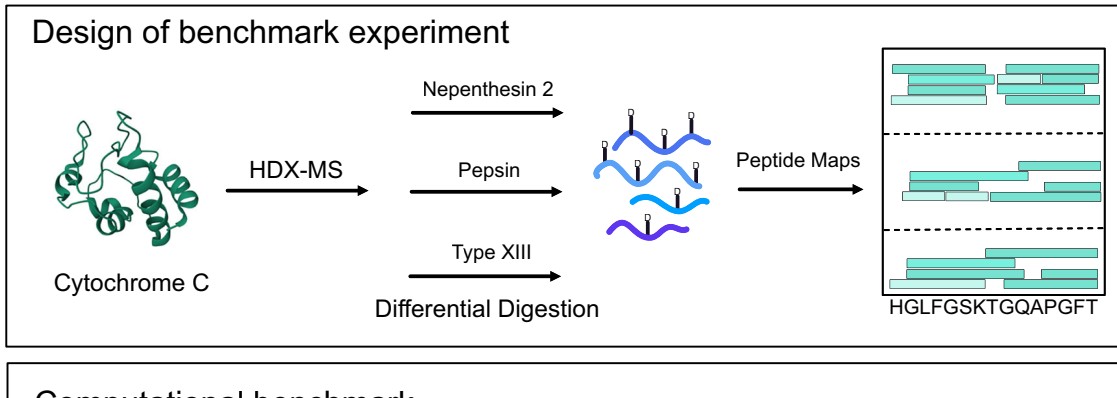

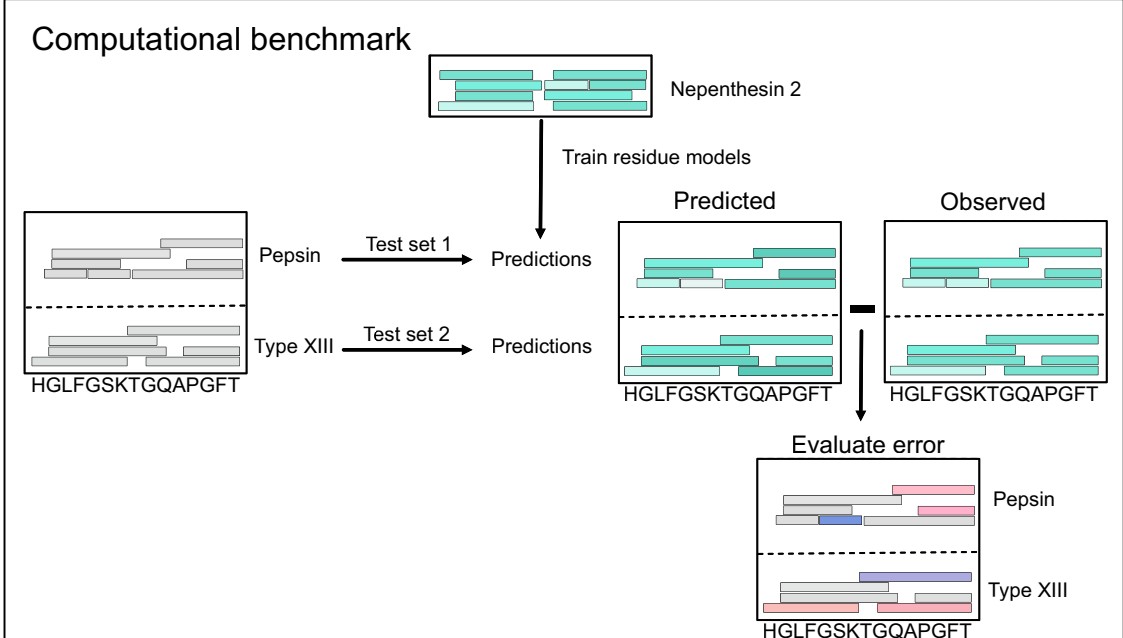

**Fig. 2 | Design of experimental and computational benchmark.** A triple-protease digest is performed on Cytochrome C to generate three different peptide maps. In turn, the data from each digest is used to train residue level models which are then evaluated on the held-out data from the other digests.

obtained from the nepenthesin II digest. Subsequently, the remaining two digests were 'held-out' and rendered invisible to the model, establishing our benchmark as a 'leave-digests-out' experiment. Each peptide from the 'held-out' data was considered in turn. We predicted their deuterium uptake by summing the already inferred residue values, thereby obtaining uptake 'peptide-maps' from both the predicted and observed data. To evaluate the performance of the model, we measured the deviation of the predicted and observed data (see Fig. 2), with the superior approaches having the lowest error. This benchmarking technique is adaptable to other digests, proteins, methods, and MS instruments, confirming its general applicability.

In order to compare to other methods, we searched the literature to find other open-source approaches. However, despite extensive treatment of the topic in the literature few methods are openly available. Only methods that were either simple enough to recode, or had available open-source code (ensuring both execution and veracity of the method) were benchmarked. Additionally, it was imperative that all benchmarked methods could report deuterium uptake, not merely differences or protection factors.

We successfully benchmarked several methods: the Pseudo-Inverse (P-Inv) approach[16], a least-squares based approach (LeastSq)[30], a weighted average approach (RFU-WA)[31], and approaches based on the Fused-LASSO (PyHDX)[26,29]. Specifically, the weighted average approach calculates a residue-weighted average, considering the length of the peptides. P-Inv inverts a coupling matrix between residue and peptide-level uptakes, while LeastSq minimizes the deviation between an uptake model and observed data using a squared error loss function. PyHDX expands on this,

incorporating an additional smoothness penalty between residues. Further details on all approaches are given in the methods.

The performance of each method under benchmark analysis is presented in Fig. 3A, revealing ReX as a consistently accurate and physically reasonable approach for inferring deuterium uptake values. Furthermore, ReX exhibited the lowest Median Absolute Deviation (MAD) consistently in six out of eight examples, outperforming other methods and only being less accurate than the generalized-inverse approach (P-Inv) at one specific time point (150 s) (see Fig. 3A). Detailed examination revealed that P-Inv, in particular, generated unphysical kinetic curves and occasionally inferred negative uptake, violating physical principles and the monotonicity assumptions of deuterium uptake (Fig. 3B). In contrast, ReX and the weighted average approach (RFU-WA) were the only methods that produced a monotonic deuterium uptake, with other methods sometimes presenting non-monotonic profiles (Fig. 3C). An extension of our benchmark analysis, which considered alternative training data and downsampled Nepthensin data, confirmed ReX's superior performance in the largest number of comparisons, outperforming the next best method, PyHDX with $\lambda = 5$, across three additional benchmarking experiments, totalling 32 comparisons (Fig. 3D and Fig. S1). In instances where PyHDX excelled, ReX was the second-best method 80% of the time, solidifying its reliability. Ultimately, the challenge of all methods is identifying the correct amount of smoothness as overly smooth approaches (RFU-WA) and those with low amounts of smoothness (P-Inv) performed poorly. We also noted that the (P-Inv) approach become unstable at points of high-deuterium uptake.

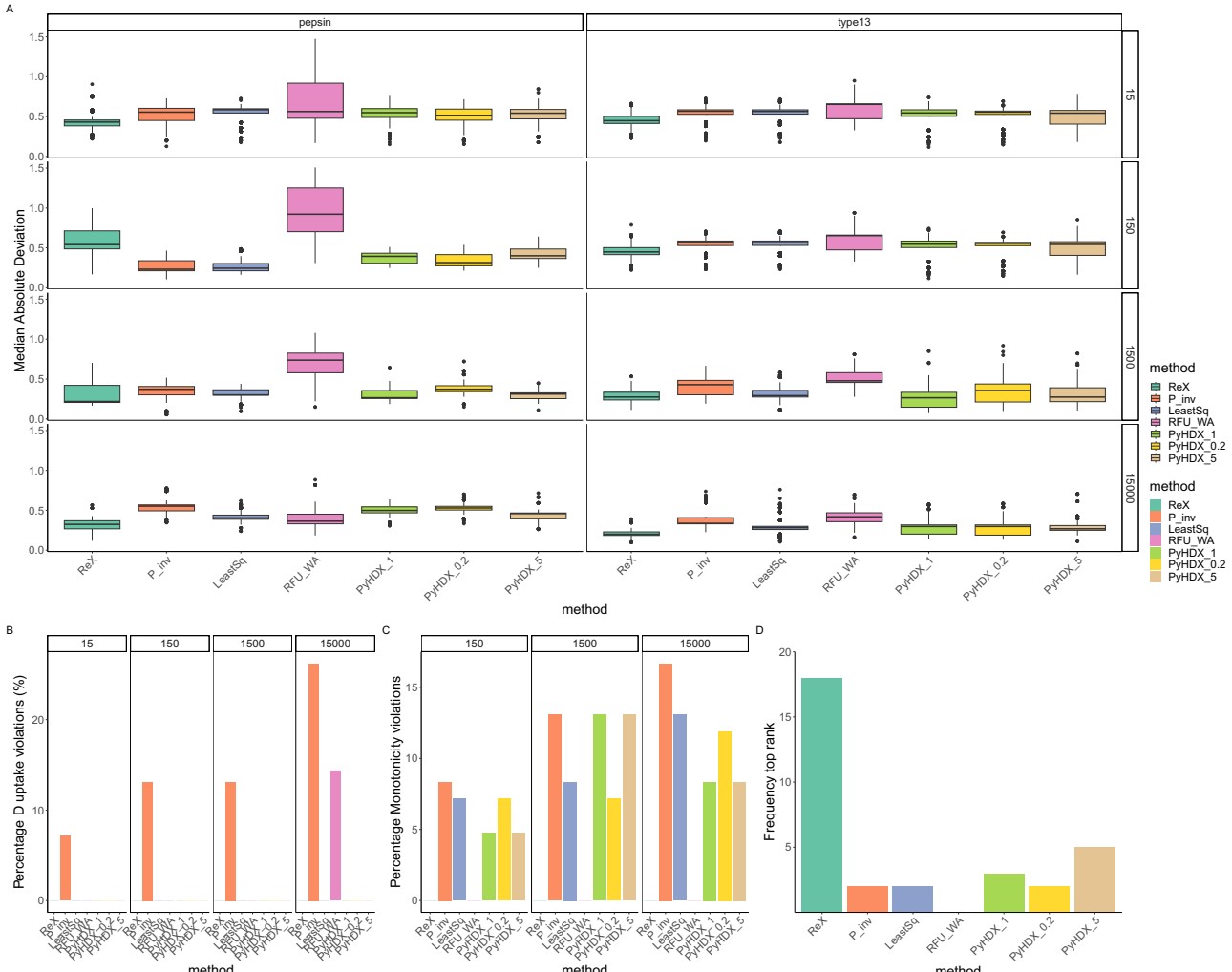

**Fig. 3 | Benchmarking results. A** The bootstrap distributions of the median absolute deviations (MAD) are reported for the held-out digests faceted by each experimental time-point. Boxplots are in the Tukey format and represent approximate 95% confidence intervals. ReX has the lowest average error in six of the eight cases. **B** The percentage of residue-level uptakes that violate uptake constraints (uptake should be between zero and one). **C** The percentage of residue-level uptakes that are monotonicity violations. Deuterium uptake should be increasing or constant over time. **D** A summary of the ranking of the methods in all benchmarking scenarios (all digests and time-points). A method is top ranked if it has lowest average error.

To our knowledge, only one dataset contains both NMR and HDX-MS measurements with fully deuterated controls[34,43]. When we applied ReX to this dataset we found contradictions between the two measurement types. In several cases, the sum of NMR residue uptakes exceeded the corresponding MS peptide uptake. This limitations arises from the size of the dataset, lack of replicates, and low redundancy rather than the modelling approach. We therefore did not include this dataset in our benchmark and instead provide a full analysis in the supplementary information (Figs. S2 and S3).

**Extracting features from ReX**

Having established the accuracy of ReX, we extracted features from the model for subsequent downstream analysis. A current limitation of the HDX-MS literature is the routine absence of reported measures of resolution of the data. ReX outputs a scale or standard deviation parameter termed sigma ($\sigma$) and uses $\sigma$ to model the deviation of its uptake fits from the observed data. Accordingly, $\sigma$ is larger for more variable data. Hence, when we added additional standard deviations of noise to our Cytochrome C data and saw that $\sigma$ increased in a close to linear fashion (see Fig. S4). This means that smaller values of $\sigma$ indicate less variable data and large values of $\sigma$ indicate greater variability. Consequently, $\sigma$ is a viable global metric for data quality and resolution.

For a residue-level measure of data quality, we propose the average reconstructive error (ARE) - a measure of the model's ability in reconstructing observed data at the peptide level (see 'Methods'). Elevated ARE values for a residue implies consistently suboptimal modelling for overlapping peptides (Fig. S4). Significant variability in the value of the ARE highlights regions where fits are either of high or low quality. Moreover, the ARE is highest in the same region of the poorly modelled peptides in our benchmarking experiment. Furthermore, these results are consistent with adding variability to the data (see Fig. S4). Crucially, $\sigma$ and the ARE are consistent, ensuring a credible interval constructed using $\sigma$ of size $1 - \alpha$ means that at least $100 \times (1 - \alpha)\%$ of the data is covered by the credible interval (see Fig. S4). Hence, $\sigma$ and ARE together are robust and communicable metrics for algorithmic faithfulness and dataset quality bringing it in-line with reporting standard in crystallography, electron microscopy and structural prediction.

We then examined four previously published HDX experiments on Alpha-Lactalbumin, Barnase, Enolase and Serum Amyloid Protein (SAP)[28], associating structural feature annotated via Uniprot[44] and Free SASA with our model outputs[45]. We defined surface accessible (SA) as relative SA above 25%[45]. Though HDX cannot simply be reduced to these quantities, we found correlations across several of these annotations. Our results (presented in Fig. 4) are obtained by randomly resampling residues 1000 times and are

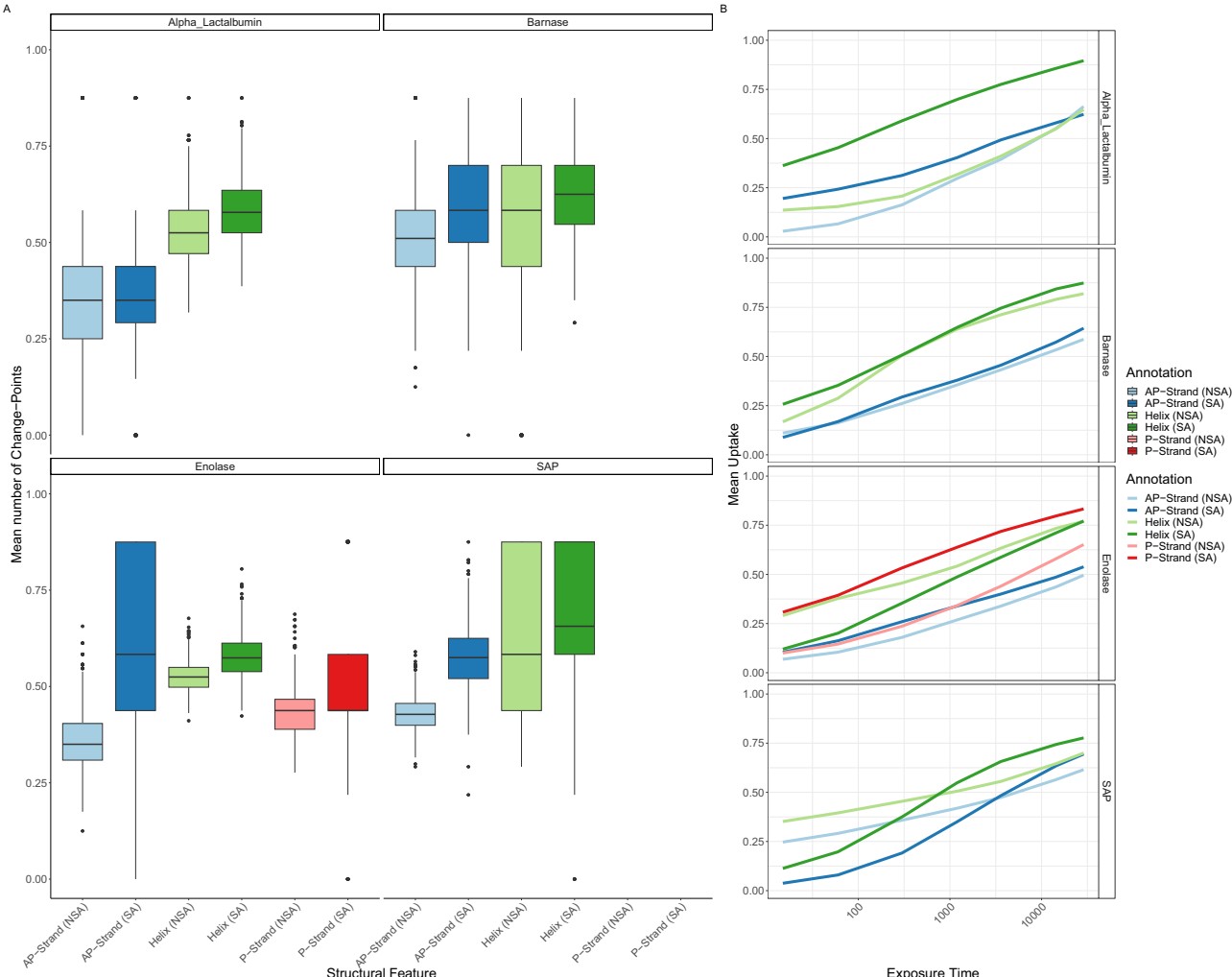

**Fig. 4 | ReX inferred features for structural annotations. A** The mean number of change points stratified by structural annotation including strand-type, helix and solvent accessibility. The distributions are over random subsamples and boxplots are in the Tukey format and represent approximate 95% confidence intervals. The greater the number of change points, the less correlated the uptake is along the sequence dimensions. **B** The same stratification as (**A**) but for the mean deuterium uptake over exposure time to heavy water. P refers to parallel and AP to anti-parallel.

reported as a distribution (see Fig. S5). Examining the mean number of change points for each of the structure features showed fewer change points for non-surface accessible (NSA) regions compared with SA regions. We interpret fewer change points (i.e. fewer jumps) as increased correlation in the sequence dimension. Hence, the result is likely because of a more stable hydrogen bond network in NSA regions of the proteins. Helices tended to have more change points than strands perhaps indicative of the helices being generally able to extend and more readily locally unfold and refold. Again, parallel (P) strands appeared less stable to exchange than their anti-parallel (AP) counterparts, perhaps due to their weaker hydrogen bond angles[46]. When considering the mean uptake of these features, we found SA regions conferred a faster rate of exchange and higher maximum uptake. The effect of weaker hydrogen bonds of parallel strands was also observed with faster and higher uptake for parallel strands compared with the anti-parallel counterparts. Helices also appeared more amenable to deuterium exchange, suggesting perhaps a propensity to breath more.

Subsequently, we used ReX's features to predict the secondary structure of predicted structures of Cytochrome C. Using AlphaFold2 and multiple sequence alignment (MSA) subsampling[47], we generated 25 models of cytochrome C (see Fig. S16 and Supplementary methods). By training random forest models with our HDX data, we achieved a median AUROC of 0.78 ([0.63, 0.88], 90% confidence interval) when predicting secondary structure (see Fig. S17 and Supplementary methods). This was higher than a

sequence-only baseline which achieved a median AUROC of 0.61 ([0.43, 0.78], 90% confidence interval). Prediction of SA and NSA were also obtained for each model and we computed the prediction confidence at each residue for both secondary structure and SA. Models with a confidence score (see Supplementary methods) above 0.7 had a median RMSD from the crystal structure of 1.2 Å, whereas models with a confidence score below 0.6 had a median RMSD of 4.1 Å (see Fig. S18). This analysis suggests HDX could potentially triage AI generated protein structures.

### Residue-resolved differential HDX-MS with ReX

HDX-MS is a powerful technique in differential settings where conformational changes induced by interactions or perturbations can be revealed[5]. Details that are often challenging or impossible to obtain with other structural approaches are readily observable using HDX-MS[48]. Having explored ReX's abilities with single proteins, we then used it to quantify residue-level difference from differential HDX-MS experiments. ReX calculates residue-level probabilities of deuterium difference, enhancing the interpretation and visualisation of the resultant data. Given that HDX alone cannot currently distinguish a binding site from conformational changes, we opt for an extensive simulation study to demonstrate ReX's abilities.

Using our already generated cytochrome C data digested by nepenthesin II, we performed 120 simulations. This involved randomly selecting a residue from cytochrome-C and then subsequent reduction of deuterium

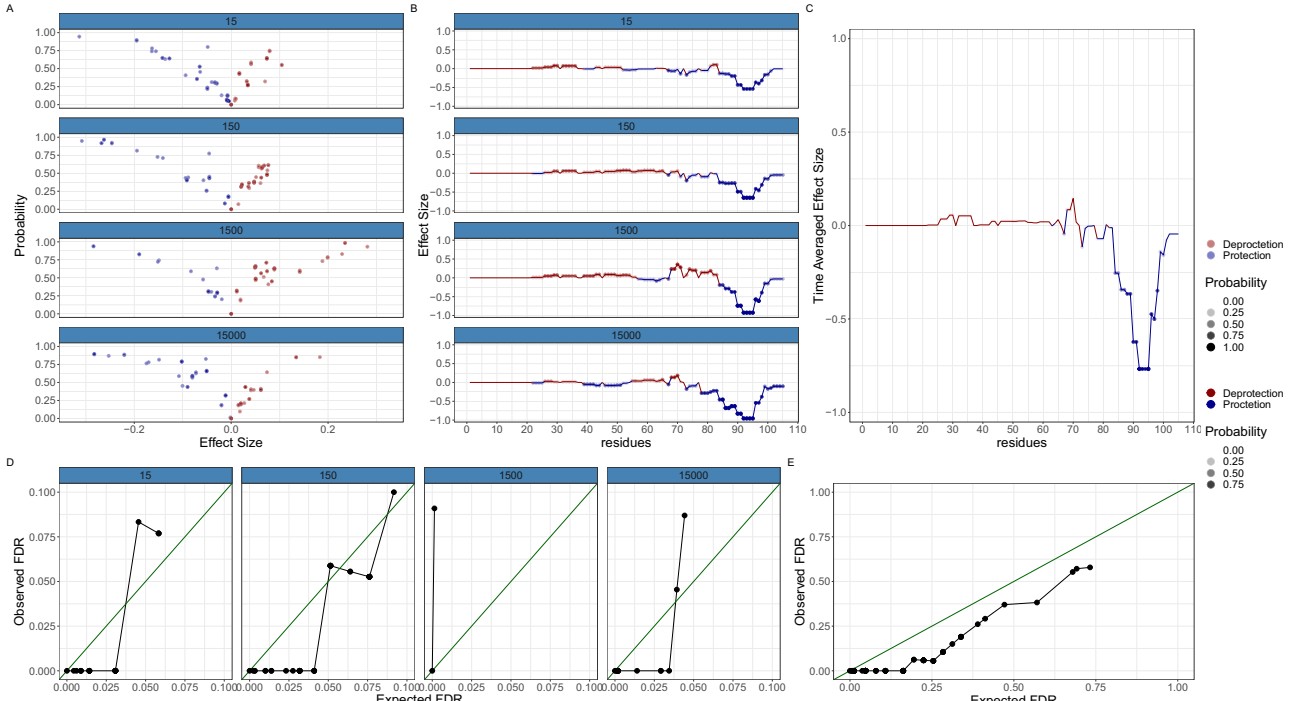

**Fig. 5 | A simulated epitope mapping example. A** Volcano plots stratified by time-points with probability of change in the y-axis and difference in Effect Size (ARE) in the x-axis. Protection is presented in blue and De-protection in red. **B** Line plots with difference in Effect Size (TRE) plotted as a function of residue. Protection is presented in blue and De-protection in red. The transparency is scaled with probability. **C** As in (**B**) but with the results averaged over time. **D** Calibration plots showing the expected false discovery rate (FDR) against the observed (FDR) are mostly conservative (points below the principal diagonal). **E** The same as (**D**) but averaged over time. The results are always conservative showing good calibration.

uptake for all peptides covering that residue to specified percentages (80, 90, 95%) of the original uptake, with 80% representing the strongest protection effect. Each scenario was repeated four times across an additional ten random seeds representing a total of 120 simulations.

The challenge of quantifying individual residue differences arises from the need to account for the consistency, or lack thereof, among overlapping peptides. ReX, through a re-constructive approach, models the residue in each protein state under study, comparing each residue's uptake with the APO (reference) state. We quantify the difference at the level of peptides by predicting the peptide-level values by summing over the posterior mean values for the appropriate residues. Subsequently, the total reconstructive error (TRE) (also effect size) is the value obtained by summing the effect for overlapping peptides. This summation approach, rather than average, gives more confidence to residues overlapping numerous peptides with a consistent direction of effect. For example, suppose four peptides overlap a residue and their deuterium difference is in random directions with consistent magnitude, then their sum will be close zero. In contrast, ten overlapping peptides each with the same direction of effect, but with small magnitude of change, will have greater value boosted by the summation of ten values. An associated probability calculation of confidence can be made (see 'Methods' for more details).

An illustrative analysis is shown in Fig. 5 for an 80% effect. Volcano plots, representing effect-size (TRE) vs Probability, are obtained for each time point with residues of interest located in the top corners (see Fig. 5A). However, volcano plots obscure spatial context, which can be re-introduced through line plots detailing effect size as a function of residue number, where transparency is scaled with probability. Our simulations induced a ripple effect where a clear protection effect is accompanied by a corresponding and adjacent de-protection effect. A subsequent protection effect but smaller is then observed proximally and so forth. When averaging the effect over time these effects are washed-out and the region of protection is clearly identified. Probabilities were transformed into an estimate of the expected false discovery rate (FDR), obtained by computing the average of the posterior error

probabilities (see 'Methods'). For typical FDRs of interest ($FDR < 0.01$), we find that our probability estimates are calibrated but can underestimate the error at higher values of FDR or at specific time-points (see Fig. 5D). Averaging over the time dimension yielded consistent estimates of error (Fig. 5E). The complete set of examples is shown in the supplementary material, which comprises one week of simulations on 48 CPU cores.

An exhaustive evaluation of all simulations is detailed in the supplementary material, where initially the proximity of the randomly selected residue (for subsequent peptide perturbation) to the residue with highest probability of difference is evaluated. Typically, these were extremely close being within a couple of residues (see Fig. S6). Furthermore, it may not always be possible to improve these results if the residue was in a region of low redundancy. We then obtained the probability of this randomly selected residue having a deuterium difference. At an 80% effect, this randomly chosen residue had close to probability 1 in almost all examined scenarios with the probability reducing (as expected) for smaller protection effects. Our analysis confirms that our approach is well-calibrated and advocates for considering temporal averages if the effects and probabilities are small. Additional calibration plots are shown in Fig. S7.

## Comparative analysis of tandem Bromodomains in BRD4

The BET (Bromo- and Extra-Terminal domain) protein family, containing epigenetic readers such as BRD2, BRD3 and BRD4[49,50], have drawn significant attention due to their role in the recognition and binding to acetylated lysine residues on histones that regulate transcription[51]. Inhibitors of the BET family of proteins have shown promise for a wide variety of therapeutic applications[52]. These proteins contain tandem Bromodomains, where the functional role of each Bromodomain BD1 and BD2 has recently been clarified[53]. Bromodomains have a characteristic four-helix bundle arranged in a left hand twist with high sequence similarity. For example, the majority of Bromodomains contain a canonical WPF shelf and a conserved asparagine (residue 140 in BD1 of BRD4). Sequence alignment of the two

Bromodomains in BRD4 reveals high levels of conservation (see Fig. S15), yet subtle differences impact the respective domain's flexibility[53].

To understand the flexibility differences at residue-level between BD1 and BD2 of BRD4, we analysed triplicate HDX-MS experiments of BRD4 in the context of the non-specific inhibitor I-BET151[54]. Five labelling time-points (15, 60, 600, 3600, 14,400s) were collected in triplicate in the presence of DMSO or I-BET151. We then applied ReX separately to each Bromo-domain due to missing data in the proline-rich disordered region joining them.

Upon examining BD1, a strong stabilisation signature was observed at all time points (Fig. 6A) suggestive of molecule binding forming additional bonds and reducing HDX. The effect endured over extended labelling times, with minimal confident de-protection, indicating global stabilisation of the entire domain (Fig. 6B). Meanwhile, BD2 presented an even more pronounced global stabilisation effect (Fig. 6C). However, I-BET151 has a 13-fold higher domain selectivity for BD1 over BD2, attributed to a through-water interaction of the quinoline nitrogen to the side chain of Gln85 in BD1[55]. To gain further insights, we examined the time averaged effect in a line plot for both Bromodomains (Fig. 6D). The results are similar with marked stabilisation around the WPF shelves. However, the BC loop in BD2, proximal to the conserved asparagine, exhibited a much stronger stabilization signature than in BD1.

To rationalise this result, we examined the structure of the small molecule in the binding pocket and saw that I-BET 151 faces away from the BC loop (Fig. 6E) meaning that it unlikely that the molecule forms any new bonds in this region. Given that the selective inhibitor IBET-BD1 achieves selectivity by additionally binding to Asp144 via an appended pyrrolidine[53], we hypothesised that the BC loop, and the whole BD2, exhibits more intrinsic flexibility. Consequently, the enhanced stabilisation signature in BD2 might be attributed more to differential flexibility than to a significant disparities in the bonding network.

## Conformational signature analysis of LXRα

LXRα is a member of the nuclear hormone receptors (NHR) family of transcription factors[56,57]. Upon binding with cholesterol derivatives, LXRα heterodimerizes with the retinoid X receptor to modulate gene expression[58]. Liver X receptors (LXRs) play a pivotal role in cholesterol homeostasis and are therapeutic targets for atherosclerotic cardiovascular disease[58,59]. Induction of cholesterol transporters, such as ABCA1, by LXRs facilitates efflux of cholesterol from macrophages and subsequent excretion of cholesterol from the body[60]. However, developing anti-atherogenic compounds has posed challenges because of their adverse lipogenic effects[61,62], a consequence of LXRs regulating the expression of many lipogenic genes[61]. Furthermore, it is believed that among LXRs it is LXRα that is primarily responsible for the lipogenic effects[61]. Two therapeutic molecules WAY-252623[63] and BMS-852927[64] have reached clinical evaluation for their attenuated lipogenic effects and potent ABCA1 induction.

In order to understand how structural dynamics correlate with the in vivo pharmacology of the different available LXR compounds, Belorusova et al.[65] conducted an HDX-MS study on seventeen molecules complexed with LXRα. These molecules were characterised by their in vivo beneficial and adverse effects using mouse models, then ranked based on their ABCA1 induction and induced plasma triglyceride (TG) levels. AZ5, AZ8, AZ9 had WAY-252623 exhibited low ABCA1 induction, with the remainder classified as being high inducers. Furthermore, AZ1, AZ876, T0901317, WAY-254011 and F1 were classed as lipogenic, AZ2-AZ5 were not classified, and the rest classified as non-lipogenic.

Given, the large number of molecules performing pair-wise comparisons was infeasible; thus we (and the previous authors) applied multivariate statistical methods. Conformational signature analysis (CSA) was performed, whereby the HDX-MS data is formatted as a data matrix and treated statistically. Unlike the previous analysis, ReX was used to project the peptide-level data to residue-level data and additionally quantify uncertainty. The uniform treatment of all states enabled examination of the conformational landscape spanned by the perturbation induced by the ligands. Subsequently, we correlated this landscape with the in vivo determined pharmacological properties of the ligands.

First, ReX was employed and followed by principal component analysis (PCA) to visualise the high-dimensional data (see 'Methods'). PCA projects the data along orthogonal components, where the first component (PC1) has highest variance explained (largest eigenvalue) and subsequent components are then ordered by the amount of variance explained (see 'Methods')[66]. The first three principal components are visualised across Fig. 7A and B, with molecules colour-coded by molecular annotations. BMS-852927 was separated from all other molecule in both PC1 and PC2. This corresponded to considerably stronger stabilisation of helix 3 and the beta-sheets, and reduced stabilisation of helix 10-12. ABCA1 inducers separated broadly along PC1, whereas lipogenic molecules, though grouping together, presented more complex interpretation. The loadings, visualised directly on the structure in Fig. 7G, indicate residues' contribution to the component variance with darker shades representing stronger contributions. As expected PC1 is driven by variation in beta-sheet and helix 3 stabilisation. PC2 variation is driven by residues on helix 3 and 10, while PC3 variation is contributed to by residues on helix 9 and the loop from helix 9 to 10.

Adopting a flexible approach, functional annotations of the molecules were used in a supervised model. We employed orthogonal partial least squares discriminant analysis (OPLS-DA[67]), creating separate models for lipogenicity and ABCA1 induction (Fig. 7C). OPLS-DA directly separates the data along a predictive component and subsequent orthogonal component(s). Associated residue weights (loadings) are depicted in the line plots in Fig. 7E and F. The separation between high and low ABCA1 inducers was driven by residues on helix 3, the adjacent beta-sheet, and helix 5. Whereas separation along the lipogenic dimension involved larger contributions from helix 1, whether stabilisation and destabilisation depended on the molecule. Notably, the lipogenic molecules saw marked stabilisation of residues 214–219 and the residues 200–214 tended to be destabilised in non-lipogenic molecules. However, these effects are partially correlated with residues that contribute to ABCA1 induction, complicating the differentiation of the competing effects. Multidimensional scaling (MDS) on chemical (Tanimoto) similarity confirmed that molecules did not cluster, in chemical space, by their functional annotations (see Fig. S10).

Providing additional context, CSA was re-performed using ReX in the context of LXRα complexed with the second receptor interacting motif of the co-activator SRC1[65]. Unliganded LXRα does not interact with the SRC1 peptide and all compounds induced binding of LXRs to the SRC1 peptide[65]. While separating the high and low ABCA1 inducers was more challenging, a clearer separation of the lipogenic classes was observed (see Supplementary Figs. S11–S13)) with (18.3%) variance explained by the predictive component (as opposed to 15.1%). The ABCA1 separation was driven by similar residues as found without the co-activator peptide, with these residues also contributing to the class separation of the lipogenic molecules, but to a lesser extent. Moreover, residues on helix 10-12 gave a high contribution, which was not observed in the ABCA1 results or without the SRC1 peptide. Results from helix 1 changed in this context, with all ligands contributing to strong stabilization of this region. Lipogenic molecules appeared to enhance the interaction with SRC1 (mean $K_d = 1.2$ μM) with a weak agonist effect from the non-lipogenic molecules[65]. In addition, the co-activator peptide interacts with helix 12 supporting the marked reduction of HD exchange. Hence, the results might be explained by lipogenic molecules encouraging higher differential recruitment of the SRC1 peptide, explaining their HD results, while ABCA1 induction seems independent of this effect. Therefore, the therapeutic effects of LXRα ligands may hinge on the conformations they stabilize. Agonist ligands encourage helix 12 folding into the active state, creating a co-activator-binding surface and recruiting SRC1[65]. Stabilisation of helix 3, 5, and corresponding beta sheets, which are the interaction sites of co-repressors induced high levels of ABCA1. Perhaps the LXR co-repressors are disassociated from the complex during these conformations leading to elevated ABCA1 levels[65].

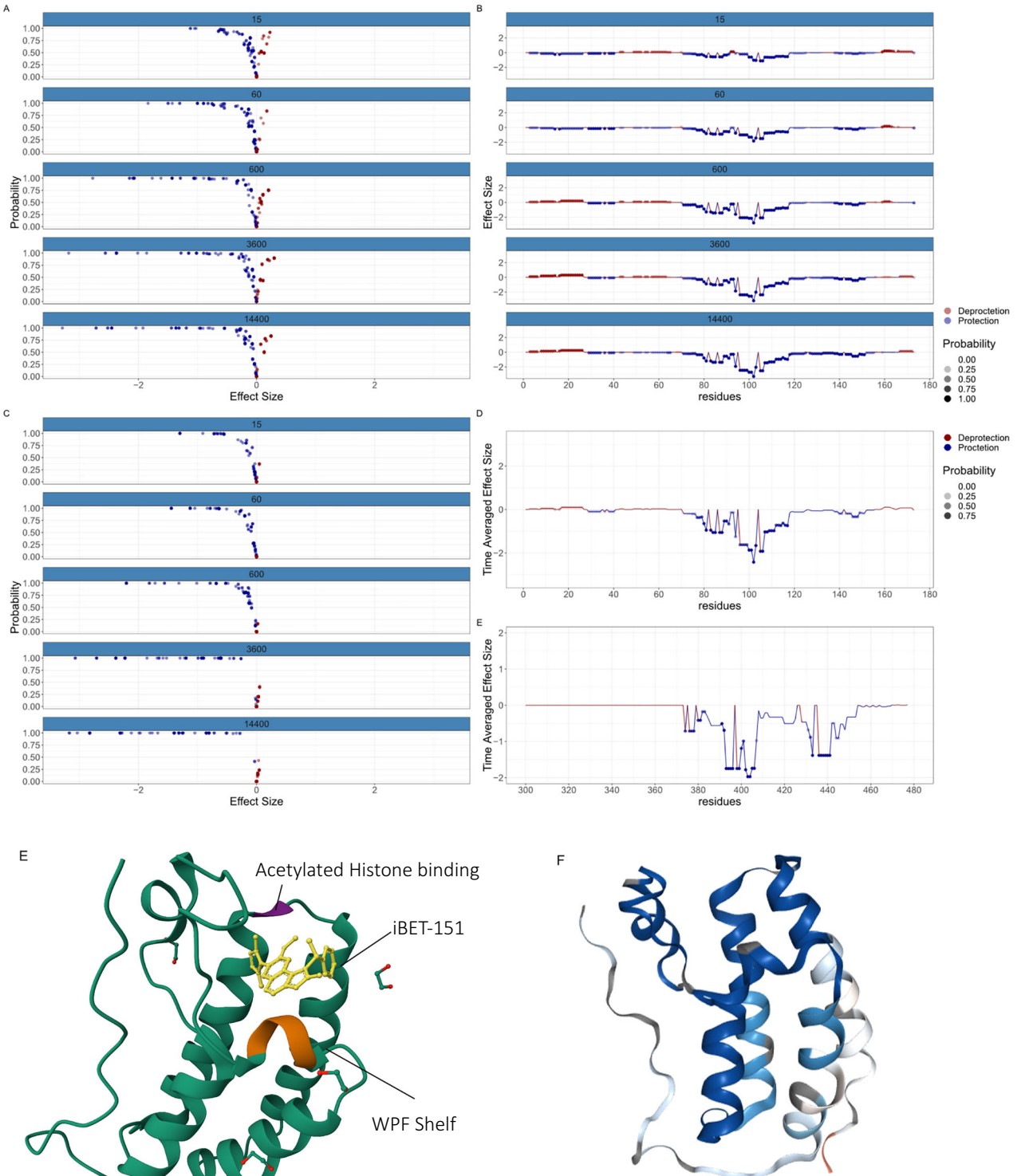

**Fig. 6 | Comparative analysis of tandem Bromodomains in BRD4 using ReX.**
**A** Volcano plots of Effect Size (TRE) against probability of difference of BD1 in
BRD4. **B** The corresponding line plots associated with BD1 in BRD4. Transparency
is scaled with probability. **C** As with (**A**) but for the BD2. **D** (Upper) A time averaged
line plot showing protection effects as a function of residue for BD1 (lower) as with

(upper) but for BD2, **E** iBET-151 in complex with BD1 of BRD4 with conserved
asparagine and WPF shelf highlighted. The BC loop faces away from the molecule.
PDB: 3ZYU. **F** A cartoon of BD1 in BRD4 with the HDX results plotted directly on
the structure. PDB: 3ZYU.

## Uncertainty quantification of CSA

In the analysis of LXRα above, we have predominantly used point estimates,
specifically means, from our Bayesian non-parametric approach ReX. While
beneficial, this approach does not exploit the full potential of the metho-
dology. Quantifying uncertainty in these analyses is paramount if they are to

replace mouse experiments or be used for downstream pharmaceutical
decisions.

To visualise the uncertainty in the conformational landscape, we project
50 samples from the posterior distribution (possible summaries) into PCA
coordinates aligned using the Procrustes transform[68,69]. Figure 8a is

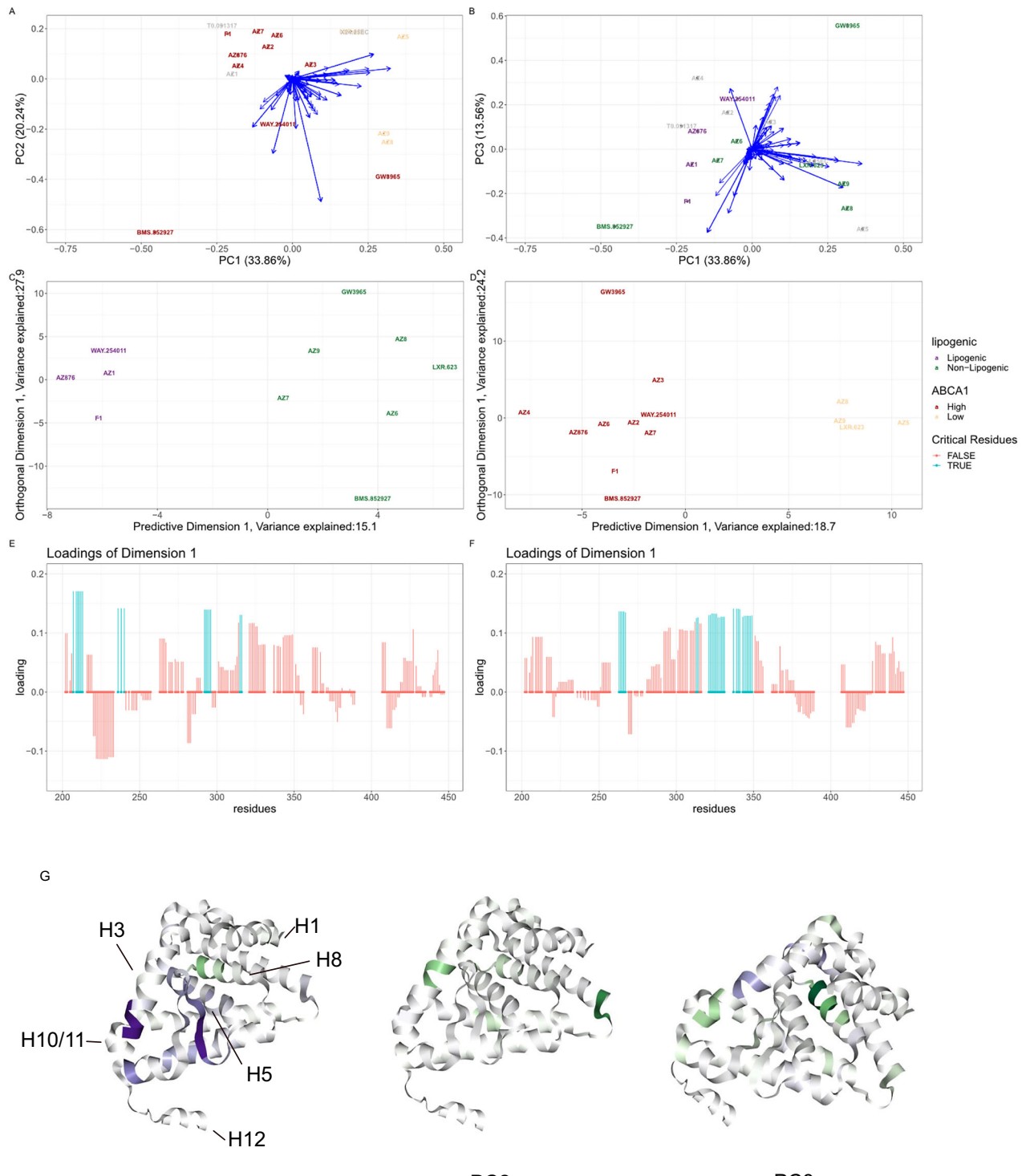

**Fig. 7 | Conformational signatures of LXRα. A** PCA plots of HDX-MS signatures per ligand. Ligands are coloured by ABCA1 induction and blue arrow represents residue contributions. **B** As with (**A**) but for PC1 versus PC3. The ligands are coloured by lipogenic annotation. **C** An OPLS-DA plot with lipogenic as the outcome variable. **D** An OPLS-DA plot with ABCA1 induction as an outcome variable. **E** The loading plot corresponding to the predictive dimension of (**C**) larger loadings indicate greater contribution. **F** A loading plot corresponding to the predictive dimension of (**D**). **G** PCA loading of (**A**) and (**B**) plotted directly on the protein structure of LXRα. Purple indicates a positive loading (positive correlation with the original variable) and Green indicates a negative loading (negative correlation with the original variable).

particularly illuminating, confirming that BMS-852927 and GW3975 are positioned in distinct regions of the plot and the resulting contours suggest that these conformations are unlikely to overlap those induced by other molecules. Other molecules are also distinct or overlap somewhat, suggesting they induced similar conformations. In fact, numerous molecules, either annotated as high ABCA1 inducers or unannotated, overlap within the same spatial region, hinting that two unannotated molecules (AZ1, T0901317) could be potent ABCA1 inducers. All lipogenic molecules (and some non-lipogenic) occupy this spatial region, indicating that avoiding this part of the conformational landscape may be key to circumventing lipogenic side effects.

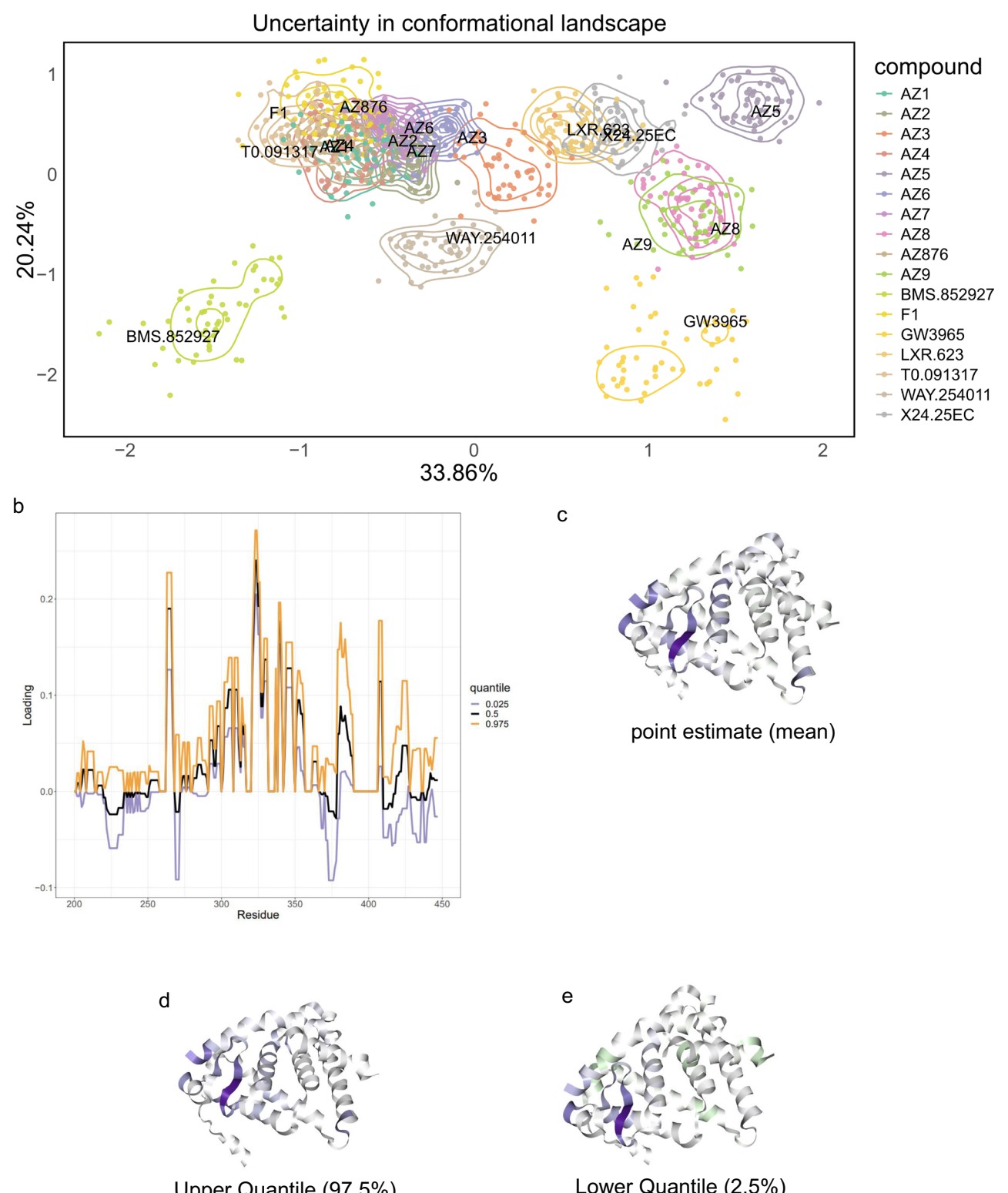

**Fig. 8 | Quantifying conformational uncertainty. a** A PCA (PC1 vs PC2) plot with uncertainty contours representing the posterior distribution of the locations of molecules in PCA coordinates. **b** The corresponding quantiles in the loadings of the principal components. **c** The loading plotted directly on the protein structure corresponding to the mean estimate of the loadings. **d** As with (**c**) but the upper quantile of the loading (**e**) as with (**d**) but with the lower quantile of the loadings.

Exploring the uncertainty in the loadings, visualized in Fig. 8b, reveals the confidence levels attributed to specific residues contributing to variation in the conformational landscape. Caution is advised when interpreting residues where the loading contribution changes sign. This does not inherently indicate a change in the direction of protection or de-protection but if the weight does intersect zero this means that we are unsure of a non-zero contribution. However, the majority of residue contributions remain confident leaving conclusions unchanged. The C-terminal helix region

displays substantial uncertainty, warranting further investigation in the context of the co-activator peptide (Supplementary Fig. S14). The quantiles, visualised in structure space and exemplified by plotting a 95% quantile directly onto the crystal structure (Fig. 8c–e), exhibit minor changes between the upper and lower quantiles, yet helix 3 and beta-sheets remain highlighted across the 95% confidence interval. Hence, uncertainty quantification strengthens our results and is a unique benefit of ReX.

## Discussion

HDX-MS offers a powerful approach to probe the conformational dynamics of proteins, yet the bottom-up approach to MS has inherent interpretative challenges stemming from peptide redundancy, length-bias, and contradictions. To address these challenges, we introduced ReX, a Bayesian non-parametric change-point model tailored to infer residue-level HDX patterns. Uniquely, ReX adapts to data quality and size, offers quantifiable measures of data resolution, and, importantly, quantifies uncertainty, distinguishing it from existing approaches. Its ability to correct length-bias, combined with a data-informed inference of smoothness along the residue dimension, sets ReX apart. As a flexible approach, ReX is able to model single proteins, differential experiments or large-scale comparisons via supervised or unsupervised conformational signature analysis.

Using a three-way differential digestion experiment, ReX was shown to be a highly accurate approach, while adhering to physical constraints of HD exchange. Furthermore, the analysis demonstrated the importance of smoothing in the residue dimension. Through a variety of case-studies, we demonstrated the insights that ReX obtained. For instance, considering BRD4 Bromodomains in the context of I-BET151, we determined the differential flexibility of the BC loop in the two Bromodomains. Further, CSA on seventeen molecules complexed with LXR$\alpha$ highlighted ReX's ability in linking conformational signatures with in vivo determined pharmacological properties. Hence, ReX offers a promising avenue to streamline further experimental investigations with additional confidence obtained from uncertainty quantification. Specifically, we found that the ligand BMS-852927 induced a distinct conformation of LXR$\alpha$ and the lack of stabilisation of helix 12, helix 3 and the beta-sheets could explain its therapeutic profile.

While ReX efficiently models centroided spectra, EX1 dynamics might necessitate a dual application of ReX, each tailored to a distinct population's centroid. Looking ahead, integrating ReX with alternative structural models, such as cryo-EM, and introducing 3D structure-derived constraints or correlations could further improve its performance.

## Methods
### Likelihood model

We first describe our proposed model as a Bayesian generative model (see ref. 32 for a general description and mass-spectrometry examples). We let $y_i(t_m)$ be the observed deuterium incorporation measurements where $i$ indexes peptides from $i = 1, \ldots, N$ and $m = 1, \ldots, M$ is an index enumerating the $M$ exposure times $t_1, \ldots, t_M$ to heavy water. We assume there is an underlying residue level process that builds $y_i(t_m)$. More specifically, the observed peptide data are considered as noisy observations of some function of the unobserved residue processes. This unobserved (or latent) residue process is specified using the following functional form:

$$\mu_r(t) = (1 - \pi_r)(1 - \exp(-b_r t^{p_r}) + \pi_r(1 - \exp(-d_r t))), \quad (1)$$

which defines the unobserved process for residue $r$, where residues are indexed $r = 1, \ldots, R$. We remark that this functional form is more elaborate than previous approaches, which is to allow a very general approach. The mixture between two competing models is defined by mixing proportions $\pi_r \in (0, 1)$. These proportions the relative preference for the two most popular models. The first component describes the stretched exponential or Weibull kinetics used by a number of studies and the second component uses the model compatible with protection factors as a single exponential per residue. The rate parameter of the Weibull model is $b_r$ and the stretch parameter is $p_r$, while the rate parameter of the exponential model is defined

as $d_r$. The form of $\mu_r(t)$ is the combination of the cumulative distributions of the Weibull and exponential distributions and hence, by definition, is constrained to the interval [0, 1]. Here, a value of 0 for $\mu_r$ means that for no protein has the amide hydrogen for residue $r$ exchanged with deuterium while 1 represents complete exchange. The mixing components are learnt from the data and so concentrate on the preferred model or an interpolation between the two. We collect the parameters using $\theta(r) = (\pi_r, p_r, b_r, d_r)$, which is a function of residues.

The peptide process is derived from the residue process as the summation of residue-level processes, using a Laplace error model because of its robustness to outliers, and hence:

$$y_i(t_m) \sim \mathcal{L}\left(\sum_{\mathcal{R}_i} \mu_r(t_m), l_i \sigma^2\right),$$
$$\mathcal{R}_i := \{r : \text{start}_i + 2 \leq r < \text{end}_i, r \neq \text{Pro}\}, \quad (2)$$

where $l_i$ is the number of exchangeable amides for peptide $i$ and $\text{start}_i$ and $\text{end}_i$ denote the start and end of $i$th-peptide. The parameter $l_i$ encodes a so-called length bias, allowing the longer peptides to be more variable (or contribute less to the overall fit). We assume that variance parameter arises from the same distribution for all residues. This assumption on $\sigma^2$ allows us to borrow power to better estimate the variance for each residue (and hence each peptide). For convenience, we write $\mu_i = \sum_{\mathcal{R}_i} \mu_r$.

### Non-parametric prior model

It is pertinent to note that at the moment our model is over-specified, by which we mean that there are more parameters than there typically observed data. However, even if $N > R$, our model may still be difficult to identify. The model becomes identifiable when the redundancy for every residue is greater than one. This over-specification is necessary to complete the model, with the issue being alleviated by careful prior modelling and quantifying uncertainty.

We employ a non-parametric prior model, which is a class of flexible prior models which allow the number of parameters in the model to be inferred using the data. The prior model is the model of the likelihood parameters; that is, the model that describes $\theta(r)$. In this way if more data is observed the model is able to adapt and introduce complexity. To reduce the effective dimension of the model we use a change-point model. Here, the parameters are specified as constant segments between points of change, where the values of the parameters, the locations of the change points and the number of change points are quantities to be inferred from the data. We note that while the prior model is piece-wise constant, when we obtain the model parameters and $\mu_r$ by, for example, averaging over the posterior distribution they are not necessarily piece-wise constant, because the location and number of change points changes with each MCMC sample.

For a more precise mathematical description of the model, we specify these change points using a *Poisson process prior* with rate $\lambda/(R-1)$. The choice of rate ensures the prior mean number of change points is $\lambda$[41]. From standard analysis of the Poisson process, the number of change points, $K$, is Poisson distributed $K \sim \text{Pois}(\lambda)$. We denote the vector of change points as $\tau = (\tau_1, \ldots, \tau_K)$, which are a priori uniformly distribution and ordered in $(1, R)$, by properties of the *Poisson process prior*. We parameterise our model so that $\lambda$ represents the prior expected number of change points rather than a rate parameter, which is more intuitive for practitioners. For larger proteins, $\lambda$ should be increased proportionally to reflect the expectation that longer sequences contain more structural transitions.

There are $K + 1$ intervals to consider between each change point and the endpoints. First, let us introduce $s_r \in \{1, \ldots, K + 1\}$ indicating to which interval the residue $r$ belongs with $s_i \leq s_j$ if $i < j$. Then if $s_r = k$ then for residue $r$ the following inequality holds: $\tau_{k-1} \leq r < \tau_k$. Then we define $\theta(r) := \theta_{s_r}$, which takes on one of $K + 1$ values in the set $\{\theta_1, \ldots, \theta_{K+1}\}$ and so it is

convenient to write the following prior function:

$$\theta(r) := \theta_{l_r} = \sum_{k=2}^{K+1} \theta_k \mathbb{I}(\tau_{k-1} \le r < \tau_k), \qquad (3)$$

from which it is obvious to see that the prior model is a piece-wise constant process. The prior model specification is complete once we specify priors for each of the individual parameters. It is convenient to index the parameters using $k$ to specify this prior structure and so we write $\theta_k = (\pi_k, p_k, b_k, d_k)$ to be the vector of parameters for residues $r$ such that $s_r = k$. We specify the following hierarchical structure

$$
\begin{aligned}
d_k &\sim \mathcal{G}(d_{shape}, d_{rate}) \\
b_k &\sim \mathcal{G}(b_{shape}, b_{rate}) \\
\pi_k &\sim \mathcal{B}(\pi_{shape_1}, \pi_{shape_2}) \\
p_k &\sim \mathcal{B}(p_{shape_1}, p_{shape_2}) \\
\log \sigma &\sim \mathcal{N}(m, v^2) \\
d_{rate} &\sim \mathcal{G}(d_\alpha, d_\beta) \\
b_{rate} &\sim \mathcal{G}(b_\alpha, b_\beta),
\end{aligned}
\qquad (4)
$$

where $\mathcal{G}$ denotes the Gamma-distribution with the shape and rate parametrisation and $\mathcal{B}$ denotes the Beta distribution with two shape parameters. This model structure involves a hyper-prior structure which allows borrowing of information between the model parameters.

### Derivation of posterior distribution

For clarity and to aid understanding the nuances of the model, we derive the posterior distribution of our model. By standard computations, using repeated application of Bayes' Theorem:

$$p(K, \tau, \theta, \sigma | y) \propto p(y | \tau, \theta, K, \sigma) p(\theta | \tau, K) p(\tau | K) p(K) p(\sigma) \qquad (5)$$

First, we recall that

$$p(K) = \text{Pois}(K; \lambda), \qquad (6)$$

and the change points $\tau$ are uniform and ordered in $(1, R)$, so

$$p(\tau | K) = \frac{K!}{(R-1)^K} \qquad (7)$$

as there are $K!$ permutations of uniforms that give the same $\tau$. Given $K$, the $\tau$ state space $\Omega_{\tau, K}$ is

$$\Omega_{\tau, K} = \left\{ \tau_1, .., \tau_k \in (1, R) : \tau_1 < \tau_2 < ... < \tau_K \right\}. \qquad (8)$$

The prior for $\theta$ is

$$
\begin{aligned}
p(\theta | \tau, K) &= p(\pi | \tau, K) p(b | \tau, K) p(d | \tau, K) p(p | \tau, K) \\
&= \prod_{k=1}^{K+1} p(\pi_k | \tau, K) p(b_k | \tau, K) p(d_k | \tau, K) p(p_k | \tau, K)
\end{aligned}
\qquad (9)
$$

with the state space $\Omega_{\theta, K+1} = ((0,1) \times (0, \infty) \times (0, \infty) \times (0,1))^{K+1}$. For the observation model, we recall that:

$$
\begin{aligned}
y_i &\sim \mathcal{L}\left(\sum_{\mathcal{R}_i} \mu_r, l_i \sigma^2\right), \\
\mu_r &= (1 - \pi_r)(1 - \exp(-b_r t^{p_r}) + \pi_r(1 - \exp(-d_r t))) \\
\mathcal{R}_i &:= \{ r : start_i + 2 \le r < end_i, r \ne \text{Pro} \},
\end{aligned}
\qquad (10)
$$

Hence, we see that

$$p(y | \tau, \theta, N, \sigma) \pi(\sigma) = \prod_{i=1}^{N} \mathcal{L}\left(y_i; \sum_{\mathcal{R}_i} \mu_r, l_i \sigma^2\right) \mathcal{N}(\log \sigma; m, v^2). \qquad (11)$$

The parameter space for fixed $K$ is $\Omega_K$, where

$$\Omega_K = \Omega_{\theta, K+1} \times \Omega_{\tau, K} \times \Omega_\sigma, \qquad (12)$$

where $\Omega_\sigma = (0, \infty)$. The full parameter space $\Omega^*$ for the posterior $\pi(K, \tau, \theta, \sigma | y)$ is

$$\Omega^* = \bigcup_{K=0}^{\infty} \bigcup_{(\tau, \theta, \sigma) \in \Omega_K} (\tau, \theta, \sigma, K). \qquad (13)$$

### Derivation of reversible-jump Markov-chain Monte-Carlo sampler

If the number of parameters in the model was fixed, we could employ a Markov-chain Monte-Carlo algorithm to sample from the posterior distribution of the model[70]. However, we are required to explore an infinite dimensional state-space or in other words our model is allowed to change dimension. That is the dimension of the model is to be inferred from the data which allows our model powerful flexibility. As a results, we require Monte-Carlo moves that allow moves between models of smaller and greater dimension. The number of parameters is essentially controlled by the number of change points. That is the increasing number of change points increases the number of parameters in the model.

This is classically referred to as a class of statistical problems where "the number of things we do not know is something we do not know". To sample from the posterior distribution of our model, we need an MCMC algorithm that allows for a varying dimension model. One such an approach is referred to as reversible jump Markov-chain Monte-Carlo (RJMCMC), sometimes referred to as trans-dimensional MCMC. We briefly introduce RJMCMC in the general setting before specialising to our case and refer to the seminal work of Green[39].

### Preliminaries on RJMCMC

Suppose that we have a countable collection of models over which we wish to perform inference $\{\mathcal{M}_K, K \in \mathcal{K}\}$. Each model $\mathcal{M}_K$ has a vector of parameters $\theta_K \in \Omega^{n_K}$, where the dimension $n_K$ is allowed to vary for each model. Bayesian model choice for observed data $y$, can be written using the following hierarchical structure:

$$p(K, \theta_K, y) = p(K) p(\theta_K | K) p(y | K, \theta_K). \qquad (14)$$

For convenience, we set $x = (K, \theta_K)$. For given $K$, $x \in \{K\} \times \Omega^{n_K} =: \Omega_K$ and $x$ varies over $\Omega^* = \bigcup_{\mathcal{K}} \Omega_K$. Now, we need to construct a Markov transition kernel $P(x, dx')$ that is aperiodic, irreducible and satisfies detailed balance. In the case of a transition between parameter subspace with different dimensions, some care is needed. Consider a Markov-chain where the current state is $x$, and propose to make a move of type $m$ that transitions to state $dx'$ with probability $q_m(x, dx')$. Not all states will be accessible from $x$ is a single move and so $q_m(x, \Omega^*) = 0$ for many $m$. Such a move is then accepted with the following probability

$$\alpha_m(x, x') = \min\left\{ 1, \frac{\pi(dx')}{\pi(dx)} \frac{q_m(x', dx)}{q_m(x, dx')} \right\}. \qquad (15)$$

However, it is still unclear how to construct moves between different models. Suppose $\rho_{K, K'}$ is the probability of proposing a move to model $K'$ when the current model is $K$. Typically, $\rho_{K, K'} = 0$ if $|K - K'| > 1$ for simplicity, but still ensures all states are accessible and therefore the Markov-

chain will be irreducible. The reversible jump MCMC algorithm can be stated as follows

- Suppose $X_t = (\theta_K, K)$ is the current state of the Markov-chain. State $X_{t+1}$ is generated as follows:
- Set $K' = K + 1$ with probability $\rho_{K,K+1}$ and otherwise let $K' = K - 1$.
- Simulate $u \sim g_{K,K'}(.)$, where $\dim(u) = K$
- Set $(\theta_{K+1}, u') = \psi(\theta_K, u)$, where $\psi$ is a differentiable bijection.
- If $K' = K + 1$, i.e. dimension increasing, we compute:

$$
\begin{aligned}
&\alpha(\theta_{K+1}, K | \theta_K, K) \\
&= \min\left\{ 1, \frac{\pi(\theta_{K+1}, K+1|y)\rho_{K+1,K}}{\pi(\theta_K, K|y)\rho_{K,K+1}g_{K,K+1}(u)} \left| \frac{\partial \psi_{K,K+1}(\theta, u)}{\partial(\theta, u)} \right| \right\}
\end{aligned}
\tag{16}
$$

- If $K' = K - 1$, i.e. dimension decreasing, we compute:

$$
\begin{aligned}
&\alpha(\theta_{K-1}, K-1 | \theta_K, K) \\
&= \min\left\{ 1, \frac{\pi(\theta_{K-1}, K-1|y)\rho_{K-1,K}g_{K-1,K}(u')}{\pi(\theta_K, K|y)\rho_{K,K+1}} \left| \frac{\partial \psi_{K,K-1}(\theta, u)}{\partial \theta} \right| \right\}
\end{aligned}
\tag{17}
$$

- Sample $\nu \sim U[0, 1]$, and if $\alpha \geq \nu$, then set $X_{t+1} = (\theta_{K'}, K')$ and otherwise set $X_{t+1} = (\theta_K, K)$

It is often easier to compute the acceptance probability by considering the probability distribution of $\theta_{K'}, K' | \theta_K, K$, which we call $Q$. The acceptance probability is then

$$
\alpha(\theta_{K'}, K' | \theta_K, K') = \min\left\{ 1, \frac{\pi(\theta_{K'}, K'|y)Q(\theta_K, K|\theta_{K'}, K')}{\pi(\theta_K, K|y)Q(\theta_{K'}, K'|\theta_K, K)} \right\}.
\tag{18}
$$

We can then perform a MCMC move between models of varying dimensions. To update the parameters for fixed model dimension $K$, then one can use a standard MCMC move such as the Metropolis-Hastings algorithm.

### Construction of RJMCMC

Here, we explicitly construct the RJMCMC sampler for our *Poisson process prior* model. We proceed by first describing the moves and then computing the transition probabilities.

**Adding a dimension.** We first consider adding a dimension with probability $\rho$. We then choose an interval to break/split by sampling $j \sim U\{1, 2, ..., K+1\}$ and let the new change point be $\tilde{\tau} \sim U(\tau_{j-1}, \tau_j)$. We now have two new intervals for which we need to generate parameters and so we simulate $\tilde{\theta}_1, \tilde{\theta}_2 \sim p(\theta)$. We then set $k' = k + 1$, $\tau' = (\tau_1, ..., \tau_{j-1}, \tilde{\tau}, \tau_j, ..., \tau_K)$ and $\theta' = (\theta_1, ..., \theta_{j-1}, \tilde{\theta}_1, \tilde{\theta}_2, \theta_{j+1}, ..., \theta_{K+1})$

**Removing a dimension.** We delete a dimension with probability $1 - \rho$. Choose a change point to delete by sampling $j \sim U\{1, .., N\}$ and sample new parameters for the merged interval from $\tilde{\theta} \sim p(\theta)$. We then set $k' = k - 1$, $\tau' = (\tau_1, ..., \tau_{j-1}, \tau_{j+1}, ..., \tau_K)$, $\theta' = (\theta_1, ..., \theta_{j-1}, \tilde{\theta}, \theta_{j+2}, ..., \theta_{K+1})$.

**Transition probabilities.** For adding dimensions, we compute the following transition probabilities, which follow by book-keeping and

independence:

$$
\begin{aligned}
&Q(\theta', \tau', K'|\theta, \tau, K) \\
&= \{\text{Probability of new dimension}\} \times \{\text{Choose interval}\} \\
&\quad \times \{\text{Choose } \tilde{\tau}\} \times \left\{ \text{Choose } \tilde{\theta}_1, \tilde{\theta}_2 \right\} \\
&= \rho \frac{1}{K+1} \frac{1}{\tau_j - \tau_{j-1}} p(\tilde{\theta}_1) p(\tilde{\theta}_2)
\end{aligned}
\tag{19}
$$

and similarly for the reverse case we see that:

$$
Q(\theta, \tau, K|\theta', \tau', K') = (1 - \rho)\frac{1}{K+1}p(\theta)
\tag{20}
$$

For removing dimensions, we compute the following transition probabilities

$$
\begin{aligned}
&Q(\theta', \tau', k'|\theta, \tau, k) \\
&= \{\text{Probability of removing a dimension}\} \\
&\quad \times \{\text{Choose a change point}\} \times \left\{ \text{Choose } \tilde{\theta} \right\} \\
&= (1 - \rho)\frac{1}{k+1}p(\tilde{\theta})
\end{aligned}
\tag{21}
$$

For the reverse situation, we note that $K' = K$, and compute

$$
Q(\theta, \tau, K|\theta', \tau', K') = \rho \frac{1}{K} \frac{1}{\tau_{j+1} - \tau_{j-1}} p(\tilde{\theta}_1) p(\tilde{\theta}_2)
\tag{22}
$$

Furthermore, we need to compute the posterior

$$
\pi(\theta', K', \tau'|y) \propto p(y|\tau', \theta', K', \sigma)\pi(\theta'|\tau', K')\pi(\tau'|K')\pi(K')
\tag{23}
$$

and the formulas are given earlier.

**MCMC moves for other parameters.** Given the current iteration of the MCMC algorithm at model $K$ (i.e. $K$ change points), were the parameters $\theta$ and $\sigma$ are updated using a random-walk Metropolis-Hastings algorithm[71].

**Algorithm initialisation.** To initialise the algorithm, we need to propose starting parameters for the model. While, in principle, this can be done randomly, we take a pragmatic approach to start near a reasonable set of parameters. We first use generalised-inversion to estimate residue-level deuterium uptakes from the peptide-level data. Generalised-inversion leads to non-physical values, so we project data outside the expected range of [0,1] onto the closest boundary. We then fit for each residue our functional model using least-squares to obtain initial parameter estimates which are subsequently smoothed using a $L_1$ trend filter[72]. This provides initial parameters for our sampler.

### Other methods

In this section we detail other possible methods for determining residue-level uptake data.

**Generalised inversion approach.** Let $U_{residue}$ be a $R \times 1$ vector of unknown residue level uptakes for a fixed exposure time. Let $U_{peptide}$ be the $N \times 1$ vector of peptide level uptakes for a fixed exposure time. Let $C$ be a coupling matrix of dimension $N \times R$, which means all entries are 0 apart from the $(i, j)$th entry which is 1 if residue $j$ has exchangeable amide for peptide $i$. Then it follows that

$$
U_{peptide} = CU_{residue}.
\tag{24}
$$

In general $C$ is rectangular is we can solve for unknown $U_{residue}$ by performing $U_{residue} = C^\dagger U_{peptide}$, where $C^\dagger$ is the generalised inverse. The main limitations of this approach are that this inversion is unstable, there is no reason why the temporal component would be consistent, uptake values are

not guaranteed to be 0 and 1 and no smoothing between uptake value between neighbouring residues.

**Least-squares approach.** Consider the notation of the following section, then $U_{residue}$ can be solved by considering the following minimisation problem

$$\min \| U_{peptide} - CU_{residue} \|_2, \tag{25}$$

constraining the entries of $U_{residue,j}$ to be in $[0, 1]$. This problem can solved by gradient-descent. This approach ensures the residue uptake values are between 0 and 1, however, all the other limitation remain. Furthermore, the minimum is not necessarily unique or global.

**Fused-LASSO approach.** To introduce smoothness between residue; that is, constraining the uptake between neighbouring residues to be similar we can introduce a fusion penalty. Consider the following optimisation problem:

$$\min \| U_{peptide} - CU_{residue} \|_2 + \lambda_{LASSO} |U_{residue,j} - U_{residue,j+1}|_1, \tag{26}$$

constraining the entries of $U_{residue,j}$ to be in $[0, 1]$. The extra penalty ensures that neighbouring residues are similar. This is the approach implemented by PyHDX[29] and is similar to ref. [26] and is part of a more general strategy called the fused LASSO. It is unclear how to select $\lambda_{LASSO}$ and because of double-dipping if $\lambda_{LASSO}$ is chosen using the data then computing $p$-value for difference after applying this approach leads to inflated false positives. This is referred to as the selective inference problem.

**Residue-averaging approach.** Here, we simply take the average deuterium uptake over peptides for the respective residues. Consider the matrix $U_{peptide}$ and normalise the columns by the number of exchangeable amides for the relevant peptide to construct $\tilde{U}_{peptide}$. The residue level uptake is then the row-average of the non-zero entries of $\tilde{U}_{peptide}$, which is a vector of length $R$. This approach suffers from the same issues as the other approaches. However, it also over smoothness because it cannot quickly adapt to sudden changes.

### ReX methodology

**Basic quantities.** In this section, we detail a number of additional quantities available from analysis with our approach other than estimate of residue level uptakes and parameters. Firstly, for each observed peptide $y_i$, we can extract the posterior mean $\hat{\mu}_i$ from our model:

$$\hat{\mu}_i = \frac{1}{M} \sum_{m=1}^{M} \mu_{im}, \tag{27}$$

where $m = 1, \ldots, M$ indexes over MCMC iterations. This is the prediction of that peptide from the model. Slight variations are of course possible and modelled by deviation from the mean using $\sigma$. We can then compute the reconstructive error (RE) using the following formula:

$$RE_i = y_i - \hat{\mu}_i, \tag{28}$$

which can be normalised by the $l_i$. To define a per residue error, we consider the collection of peptide $N_r$ overlapping residue $r$. We then defined the (signed) average reconstructive error (ARE) and the total reconstructive error (TRE) as follows:

$$\text{Signed ARE}_r = \frac{1}{n_r} \sum_{j \in N_r} \frac{y_j - \hat{\mu}_j}{l_j},$$

$$\text{ARE}_r = \frac{1}{n_r} \sum_{j \in N_r} \frac{|y_j - \hat{\mu}_j|}{l_j}, \tag{29}$$

$$\text{TRE}_r = \sum_{j \in N_r} \frac{y_j - \hat{\mu}_j}{l_j},$$

where $n_r$ is the redundancy at residue $r$. We can see that any of these quantities would be large if the data are poorly modelled at residue $r$ and these quantities give us an idea of which residues are contributing most to the error. $\text{TRE}_r$ is particularly important in the context of HDX-MS because if several peptides overlapping a residue all with the same direction of error/difference, such a difference is then consistent. This is a natural way to exploit the redundancy of the data.

**Probabilistic estimates.** We note that $\sigma$ is an estimate of the deviation at any residue $r$. Therefore, we may compute the following probabilities:

$$P(\text{Signed ARE}_r > 0) \approx \frac{1}{m} \sum \mathbb{I}(\text{Signed ARE}_r > \tau^{(m)}), \tau^{(m)} \sim \mathcal{L}(0, \sigma^{(m)})$$

$$P(\text{ARE}_r > 0) \approx \frac{1}{m} \sum \mathbb{I}(\text{ARE}_r > |\tau^{(m)}|), \tau^{(m)} \sim \mathcal{L}(0, \sigma^{(m)})$$

$$P(|\text{TRE}_r| > 0) \approx \frac{1}{m} \sum \mathbb{I}(|\text{TRE}_r| > |\tau^{(m)}|), \tau^{(m)} \sim \sum_{N_r} \mathcal{L}(0, \sigma^{(m)})$$

$$\tag{30}$$

These values give us an estimate of errors at a particular residue. In practice, we may wish to control the false discovery rate (FDR) but given that in a typical application we do not know the false positives, we have to estimate this quantity. We can estimate the expected false discovery rate (EFDR), in the following manner. Consider a set of ordered probabilities for a particular phenomena such as deuterium difference $p_1, \ldots, p_n$. For a particular threshold $\tau$, we estimate the EFDR as follows:

$$\text{EFDR}(\tau) = \frac{\sum_{i=1}^{n}(1 - p_i)\mathbb{I}(p_i \geq \tau)}{\sum_{i=1}^{n} \mathbb{I}(p_i \geq \tau)} \tag{31}$$

Typically, we wish to control the EFDR at a particular level, say 0.05, however, it is not clear which threshold corresponds to this value. To do this, we simply take the smallest $p_i$ such that the following holds: $\text{EFDR}(p) < 0.05$ for $p > p_i$ and $\text{EFDR}(p) > 0.05$ for $p < p_i$. It is easy to find this value by trying sequentially smaller values of the observed $p_i$.

**Differential analysis.** HDX-MS is frequently applied in the differential setting where by two proteins states are compared. Examples include epitope mapping using an antigen with and without antibody or proteins with ligands, modifications, mutations etc. To perform differential analysis, we fit our model on the differential state being considered, for example the antigen-antibody complex data. We then using a reconstructive approach by computing the difference between the peptides observed in the APO data but using the posterior quantities from the state of interest. Large and consistent differences indicate confident deuterium difference between the two states. By using $\text{TRE}_r$ and the associated probabilities of a difference, we can quantify the residues which are different between the two conditions. The comparison need not be the APO data but any protein state of interest. It is important to note that using this approach explicitly makes use of the redundancy of the data. Consistent differences in the same direction on an overlapping group of peptides will be assigned greater confidence because the quantities are summed rather than averaged.

**Conformational signature analysis.** At a high-level, conformational signature analysis (CSA) is for applications where pair-wise comparison between all protein states is unfeasible. Typically, a large-panel of small molecules or antibodies may be compared using HDX-MS. Several goals can be achieved by employing CSA. Firstly, we wish to quantify the conformational landscape; that is, what are the modes of variation that the perturbations to the protein induce. In what ways are perturbations similar in the variation they induce to the protein. Which residues are the largest contributors to this variation? If the perturbations have associated

functional annotations (such as the molecular property of a ligand) can these be discriminated and can we quantify the residue associations? Given the complexity of these analysis, quantifying uncertainty is of importance to avoid spurious associations.

ReX provides functionality to answer these questions. We first employ ReX to project peptide-level to residue-level data, using $TRE_r$ for all states $c = 1, \ldots, C$ under consideration. These data are then combined into a single data matrix of conditions by residues $X$. That is the $(c, r)$th entry of $X$ is the $TRE_r$ for state $c$. For unsupervised analysis (data without external annotations), we employ principal components analysis (PCA). This projects the matrix $X$ using an orthogonal linear transform into a new space. The directions taken for each subsequent component explain the greatest variation of the data while remaining orthogonal. This projection can be written as following

$$S = XL, \tag{32}$$

where $S$ is the $C \times R$ score matrix and $L$ is the $R \times R$ loading matrix. Typically, the data is visualised using a 2D scatter plot using the coordinates described by $S$. The first column of $S$ describes the first principal component (PC) and so on. Dimensionality reduction is typically achieved by only keeping a small number of the columns of $S$, which explain a large amount of the variation in the data. $L$ is the $R \times R$ loading matrix which is the projection matrix in this case. Taking columns of $L$ and scaling by the variances, called loadings, define the important contributing variables to the variation quantified by the PCs. In this case, the loadings are associated with residues and the magnitude (whether positive or negative) indicate the importance of that residue. The loadings for each PC can be visualised directly on the protein structure to aid interpretation. Note that a negative or positive loading does not necessarily indicate protection or de-protection.

If each condition is associated with an external annotation such as a molecular property of a compound, we can add an annotation vector $y$. In this case, we an employ a discriminate analysis (DA) method. Similarly to PCA, we find the directions of the data that most discriminate the molecular properties using partial least squares discriminate analysis (PLS-DA) or the orthogonal extensions (OPLS-DA)[67]. In the latter, the dimensions are separated into predictive and orthogonal dimensions. The annotation vector can be categorical (antagonist vs agonist) or continuous ($EC_{50}$). As before, the loadings can be projected onto the protein structure to interpret the results.

To perform uncertainty quantification, we notice that we began our analysis with using the posterior means from the ReX output. Instead, we can repeat the aforementioned analysis using samples from the posterior distribution. In this way, we build up a distribution of locations for each projection of our state and a distribution of loadings. We align the different projections to the projection from the posterior means using the Procrustes transform.

### RexMS R-package
The software package is implemented in R and can be found at the following url: https://github.com/ococrook/RexMS. Documentation is found within the package and additional tutorials are found within the package website https://ococrook.github.io/RexMS/. The tutorial cover single protein analysis https://ococrook.github.io/RexMS/articles/ReX.html, differential analysis https://ococrook.github.io/RexMS/articles/DifferentialRexMS.html, and conformational signature analysis https://ococrook.github.io/RexMS/articles/ConformationalSignatureAnalysis.html. Conformational signature analysis is implemented in both unsupervised and supervised mode, in addition, the supervised mode allows both categorical (e.g. a functional label) and continuous (e.g. binding affinity). There can be missing values in the labels too if you wish to infer outcome given the conformational signature.

### Benchmark experiment of HDX-MS of Cytochrome C
Horse heart cytochrome c (Sigma C7752) was prepared at a stock concentration of 80 µM in an equilibrium buffer of 50 mM MOPS, 150 mM NaCl, pH 7.2. For peptide mapping experiments, 6 µL of 6.67 µM

cytochrome c (40 pmol) was diluted in 54 µL equilibrium buffer. For labelling experiments 6 µL of 3.33 µM (20 pmol) cytochrome c in 54 µL labelling buffer (50 mM MOPS, 150 mM NaCl, pD7.2 (pHread = 6.8)). The samples were labelled at 15, 150, 1500 and 15,000 s at 20 °C, performed in triplicate for experiments with nepenthesin-2 digestion, and in singlicate for aspergillopepsin and pepsin digestions. The reaction was quenched in an equal volume of quench buffer (400 mM sodium phosphate, 6 M guanidine hydrochloride pH 2.2, 2% formic acid) for 1 min at 0.5 °C and injected into a 100 µL sample loop and digested on one of three protease columns: nepenthesin-2 (2.1 mm × 20 mm, Affipro), aspergillopepsin (2.1 × 20 mm, AffiPro) or pepsin (2.1 × 30 mm, Waters Enzymate BEH).

For 100% D labelling experiments (fully deuterated control), the reaction was performed offline, where 6 µL of 6.67 µM cytochrome c (40 pmol) was diluted offline in 54 µL of 6 M d5-guanidinium deuteriochloride prepared in MOPS labelling buffer (50 mM MOPS, 150 mM NaCl, pD7.2 (pHread = 6.8)) and labelled for 15 min at 20 °C. The reaction was then quenched by manual addition of an equal volume of 400 mM sodium phosphate, pH 2.2, 2% formic acid for 1 min at 0.5 °C prior to sample injection.

The resultant peptides were collected on a pre-column trap (UPLC BEH C18 Vanguard, Waters) for 4 min with 0.2% formic acid and 0.03% TFA in $H_2O$ at a flow rate of 100 µL/min, and separated by liquid chromatography (1.7 µm UPLC BEH C18 column, 1.0 × 50 mm dimensions, 130 Å pore size, Waters) for 12 min at a flow rate of 20 µL/min at 0.5 °C, over a gradient of 11–40% of 0.2% formic acid in MeCN before ramping to 98% for a further 3 min. A Leucine Enkephalin solution was used as lock mass.

Data was acquired on a Waters Cyclic IMS series mass spectrometer. The instrument was operated in positive mode and resolution mode (V-mode), with a capillary voltage of 3.0 kV, and sample cone voltage of 20 V. The cyclic ion mobility separation was completed in a single pass using an ADC start delay of 13 ms and two pushes per bin, and a TW static height of 22 V, with sequence phases of 10 ms inject, 3 ms separate, and 34 ms eject/acquire. HDMSE mode was enabled for data-independent acquisition for the peptide mapping experiments. The spectra were collected between 50 and 2000 $m/z$, with transfer collision energy ramp from 18 to 50 V. For labelling experiments, HDMS mode was used, with spectra acquired between 400 and 2000 $m/z$.

Peptide mapping experiment data was processed in ProteinLynx Global Server v3.0.2. Identified peptides were first filtered using a precursor maximum error of ±7 ppm, a PLGS score of 7, and sequence length of 5–20 amino acids. A second LARS-type filtering system was then employed to further triage peptides, where a minimum LARS score of 5 was set based on seven soft filtering thresholds: a minimum intensity 10,000, minimum products of 6, minimum products per amino acid of 0.5, minimum consecutive products of 1, minimum sum intensity of identified products of 2000, minimum peptide score of 8, as well as an identification in ≥2 peptide mapping experiments performed. HDX-MS labelling experiments were processed and manually curated in HDExaminer (v3.3) against the filtered peptide list to determine deuterium uptakes.

To correct between uptake difference between digest experiments, even after normalisation by a full deuterated control, a correction factor was obtained by taking the peptides that were common in both experiments and computing the ratio of the medians of all these common peptides per time point. Deuterium uptakes for all peptides were then multiplicatively adjusted by this correction factor per time point to put them on the same scale.

### Data availability
Cytochrome C datasets are provided as supplementary datasets. HDX data for Enolase, Alpha Lactalbumin, SAP and Barnase are available from ref. 28. BRD4 data is available from ref. 73. LXR data is available from ref. 65.

### Code availability
The implementation of RexMS is provided open source at https://github.com/ococrook/RexMS and indexed with https://doi.org/10.5281/zenodo.17141717.

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

## Acknowledgements

We thank Geoff Nichols for useful suggestions on the modelling. The GPT-4 and Claude Sonnet 4 API was used for language suggestions. O.M.C. acknowledges funding from a Todd-Bird Junior Research Fellowship and MRC Fellowship MR/Y010078/1.

## Author contributions

O.M.C. developed the method, wrote the code, analysed the data and wrote the manuscript. N.G. generated the HDX-MS data. C-w.C. and C.M.D. edited the manuscript and supervised the project.

## Competing interests

C-w.C. and N.G. are employees of GSK. C.M.D. and O.M.C. declare no competing interests.
