## [Transparent Peer Review file · Communications Chemistry]

Inferring residue level hydrogen deuterium exchange with ReX

Corresponding Author: Dr Oliver Crook

Version 0:

Reviewer comments:

Reviewer #1

(Remarks to the Author)

This paper develops a new method to infer residue-level uptake values during hydrogen exchange mass spectrometry experiments by using a non-parametric Bayesian model. The paper compares the newly developed method to the ones available in literature and tested it using different experimental data. Results are exciting for people working in the field since residue level HDX has been difficult to achieve experimentally and available methods to infer values are not very accurate. Overall, I think the paper is suitable for publication in Nature Comms after a minor revision is performed,

General comments:

- Single residue HDX has been measured using NMR before. I suggest the authors to compare protection factors from NMR experiments to the ones obtained by Rex and evaluate their similarity.
- Rex relies on overlapping peptides to infer residue level values. Please comment on the minimum redundancy needed to have confident results.
- Please clarify what data is being simulated for Figure 5. What are the expected results there? Do the regions showing protection correlate with expected results?
- Residue resolved HDX, shows Rex being able to show differences when a 20% reduction in deuterium uptake is simulated. Can you comment on what is the minimum deuterium uptake reduction Rex needs to be able to identify the change? Most allosteric effects in proteins show minimal HDX change.

Minor comments:

Page 10, 3rd paragraph: Change Serium Amyloid Protein for Serum Amyloid Protein.

Page 10, 3rd paragraph: "being able to concertina"?

Page 10, 4th paragraph: 25 models of cytochrome C refers to S12 not S11.

Page 11, 1st paragraph: AUROC refers to S13 not S12.

Check all Supp Figures numbering.

Figure 4. Add meaning of AP,P strands both in text and in figure legend.

If possible, change the colouring of figure8, its hard to identify dots.

Methods section: Analytical column used Waters 130Å

Methods section: Please include detailed ionisation and IM settings used in the cIMS instrument

Reviewer #2

(Remarks to the Author)

Communications Chemistry

Manuscript ID: COMMSCHEM-24-0251-T

Title: Inferring residue level hydrogen deuterium exchange with ReX

Authors: Oliver M. Crook, Nathan Gittens, Chun-wa Chung, and Charlotte M. Deane

In this manuscript, the authors describe a new method for inferring residue level uptake patterns by leveraging peptide overlap, temporal data, and sequence correlation. ReX treats HDX-MS as a multiple change-point problem within a Bayesian non-parametric framework, allowing for parameter inference, differential HDX confidence assessments, and uncertainty estimation. The paper introduces a novel

computational approach that addresses significant challenges in interpreting HDX-MS data. The authors use the method to analyze the differential flexibility of BRD4's Bromodomains and to quantify conformational variations in LXR α induced by various small molecules.

The ability to infer residue-level HDX patterns with higher resolution and confidence is a valuable advancement. This will interest researchers in structural biology, particularly those studying protein dynamics and interactions. Overall, the paper is well-written and provides sufficient detail for reproducibility. ReX's ability to adapt to data quality and quantify uncertainty sets it apart from existing methods. I recommend the paper for publication, subject to the authors addressing the following point:

1) Information on the computational efficiency and scalability of ReX, particularly for large protein datasets, would be valuable. Specifically, could the authors comment on the trade-off between the accuracy of change-point detection and the computational speed of the algorithm? Highly accurate methods may be slower, while faster methods may sacrifice some accuracy.

Reviewer #3

(Remarks to the Author)

Summary

The paper by Crook *et al.* introduces and benchmarks ReX, an approach to analyze hydrogen-deuterium exchange (HDX) data using a Bayesian non-parametric model. The paper describes the probabilistic model and a Markov chain Monte Carlo (MCMC) algorithm to sample all model parameters. The probabilistic model generates peptide-level data from a representation of the polypeptide chain at a residue level. The residue-level parameters are combined via a latent change-point model. The number of change points is unknown and inferred from the data using a trans-dimensional MCMC algorithm. ReX is benchmarked on eight protein datasets and compared to other methods. Various analyses illustrate the strengths of the ReX approach.

Assessment

The paper by Crook *et al.* is innovative and statistically sound. The authors show convincingly that their approach is conceptually cleaner and performs better than existing approaches to analyze HDX-MS data. Therefore, in principle, the article should be considered for publication. However, there are various issues with the current version that should be addressed by the authors.

Major issues

Probabilistic model and inference method

The description of the model (section 4) should be improved.

1. Being rather unfamiliar with HDX-MS data, it took me some time to understand the role of time t . The exposition of the model would be clearer, if you introduced exposure times t_m such that the data are $y_i(t_m)$ and i is an index enumerating the peptides, m an index enumerating different exposure times. Moreover, it would be useful to make the time dependence of μ_r explicit in Eq. (1).

2. What is the physical justification for tying the residue-level parameters together via the change-point model? Does it mean that local segments have similar structural properties that affect hydrogen-deuterium exchange such as secondary structure and/or solvent accessibility? Shouldn't the τ_k be located at residues where these properties change significantly? Moreover, if the change points are roughly positioned where the local structure changes, shouldn't the number of change points K increase with the number of residues R (which contradicts your choice $\lambda/(R-1)$ for the Poisson rate resulting in a Poisson-distributed number of segments K independent of R)?

3. Symbols d_r, q_r are introduced in Eq. (1) and in the following explanation, but are not explained properly (perhaps they are the same?).

4. What are $\text{start}_i, \text{end}_i$ introduced in Eq. (2)? I guess these are the start and end of the i -th peptide, but the paper does not explicitly explain this.

5. I find your notation for the change-point model difficult to comprehend. One reason is that you use the symbol θ for both residue and segment-specific parameters. Why not introduce a label $\ell_r \in \{1, \dots, K\}$ for each residue r indicating to which of the K segments residue r belongs, or equivalently binary variables $Z_{rk} \in \{0, 1\}, \sum_k Z_{rk} = 1$ similar to the allocation parameters of a mixture model that would replace the indicator function $\mathbb{1}(\tau_{k-1} \leq r < \tau_k)$ in Eq. (3)? The residue-level parameters would then be obtained from the segment-specific parameters θ_k as θ_{ℓ_r} or $\sum_k Z_{rk} \theta_k$ replacing $\theta(r)$, and the likelihood could be

formulated in terms of π_{ℓ_r} , b_{ℓ_r} , etc.

6. Distributions \mathcal{G} (Gamma), \mathcal{B} (Beta), etc. in Eq. (4) are not introduced.

7. Weibull kinetics: The standard Weibull distribution allows for shape parameters p_r that are larger than 1. Why do you restrict p_r to $[0, 1]$ by the choice of a Beta prior for p_r ? Moreover, it would have been helpful (at least to me) to understand the model better, if you explained that your model of $\mu_r(t)$ combines two cumulative distribution functions (of the Weibull and the exponential distribution) and that, by construction, $\mu_r(t) \in [0, 1]$.

8. What is the motivation for using a Laplace distribution in the likelihood?

9. I would be interested in more details about the MCMC algorithm and its performance. For example, how sensitive are the results to the particular choice of hyperparameters (α , β , m , v , etc.) in Eq. (4)? What are typical computation times?

10. It can be tricky to set up reversible jump MCMC (RJMCMC). How do you ensure convergence of your MCMC sampler? How do you choose the initial parameters from which the Markov chain is started?

11. Your description of RJMCMC on page 27/28 you seem to suddenly use lowercase letters k to indicate the number of change points. I find this confusing. Moreover, you seem to always change the dimension of the model, because you only mention moves $k \rightarrow k+1$ and $k \rightarrow k-1$. Why are there no moves preserving the number of model parameters?

Other analyses

* Validation of AlphaFold models: At the end of section 2.3 you suggest that ReX could be used to validate 3D structure predictions from AlphaFold. In particular, your confidence score seems to reliably pick out regions that have a high/poor model quality, which is nice. However, have you tried secondary structure prediction from sequence only to check if the HDX-MS data are providing information that goes beyond standard secondary structure prediction?

Minor issues

Typos / grammar lapses

* General: "change points" not followed by a noun (such as in "change-point model") should not be hyphenated

* Page 10: "adding additional" sounds a bit redundant;

* Page 12: "The greater the number of change points less correlated"

* Page 13: replace "de-convolute" with "deconvolve" or simply "distinguish"?

* Page 14: "to improve this results", "our analysis ... advocate"

* Page 21: "procrusties transform" should be "Procrustes transform", "the majority of residues contributions"

* Page 24: "this functional form is more elaborate that previous approaches"

* Page 25: "The choice of rate ensure", "From standard analysis of Poisson process" ("the" missing).

* Page 25: I find the following sentence a bit cryptic:

"We note that whilst the prior model is hence piece-wise constant the posterior distribution (the distribution of the parameters having observed the data) is not necessarily in the same model family."

I guess you are saying the following: The model parameters and μ_r obtained by, for example, averaging over the posterior distribution are not necessarily piece-wise constant, because the location and number of change points changes with each MCMC sample.

* Page 28: "Contrusion of RJMCMC", "Probability of new dimesnion"

* Page 29: "we the parameters θ and σ "

* Page 31: "which residue are contributing"

Figures

* Not all figures seem to be useful. For example, figure 2 simply illustrates the fact that the benchmark is based on predicting an entire digest that was held-out during model inference. The space could be used, for example, to show more details of the actual inference such as MCMC parameter traces, marginal distributions, etc.

* Figure 1 uses the symbols ϑ_1 , ϑ_2 , etc. whereas the exposition in section 4 uses θ_k or $\theta(r)$.

* Figure S13: Panels in bottom row show model 17 multiple times, whereas the other models are only shown once.

Other minor issues

* Two meanings of λ : LASSO parameter and rate of the Poisson process that governs the change-point model.

* You do sampling. What's final prediction? A posterior mean over different change-point models?

* What are the PDB codes of the structures shown in Figure 6?

Reviewer #4

(Remarks to the Author)

I co-reviewed this manuscript with one of the reviewers who provided the listed reports. This is part of the Communications Chemistry initiative to facilitate training in peer review and to provide appropriate recognition for Early Career Researchers who co-review manuscripts.

Version 1:

Reviewer comments:

Reviewer #2

(Remarks to the Author)

Thank you for providing a clear and thorough response to my query. Your plans to re-implement performance-critical components in C++ sound promising and might be critical for the wider adoption of the method. At this stage, I have no further questions or concerns.

Reviewer #4

(Remarks to the Author)

The authors have addressed most of my concerns.

I believe it's important for readers to know that there is only one dataset containing both NMR and HDX-MS measurements with fully deuterated controls, and that RexMS produces contradictory results due to the quality of the experimental data. Please briefly address that in the main text and refer to the full analysis in the peer review/supporting information file.

With that clarification, I believe the paper is ready for publication.

We thank all the reviewers for their comments and time taken to evaluate our manuscript. Please see a point-by-point response to each comment.

Reviewer #1 (Remarks to the Author):

This paper develops a new method to infer residue-level uptake values during hydrogen exchange mass spectrometry experiments by using a non-parametric Bayesian model. The paper compares the newly developed method to the ones available in literature and tested it using different experimental data. Results are exciting for people working in the field since residue level HDX has been difficult to achieve experimentally and available methods to infer values are not very accurate.

Overall, I think the paper is suitable for publication in Nature Comms after a minor revision is performed,

We thank the reviewer for their positive review of our manuscript. We appreciate the reviewers' excitement and value their comments below which we have addressed in a point by point response.

General comments:#

-Single residue HDX has been measured using NMR before. I suggest the authors to compare protection factors from NMR experiments to the ones obtained by Rex and evaluate their similarity.

This is a valuable point raised by the reviewer. We conducted a comprehensive literature search for datasets containing both NMR and MS HDX measurements with fully deuterated controls, identifying only one suitable example (1) which was previously analyzed in (2). This is a small dataset containing only 14 peptide measurements for MS data and 27 residues for NMR data on a 101 residue mouse prion protein under the same conditions. Neither the MS nor NMR measurements were replicated and so we were uncertain to the level of error in the data. Furthermore, the average redundancy was 1 and the maximum was 3. However, for the residues that had redundancy 3 there was no NMR data available.

We performed quality control on the datasets and found that 10% of the NMR measurements were incompatible with the MS measurements. Explicitly, we observed that the sum of the uptakes for the NMR residue-level data was greater than the peptide-level MS measurements even when only summing a subset of residues that the peptide covered implying the remaining residues with no NMR measurements would have to have negative uptake measurements to be compatible with the MS data.

Stofella et al. showed that the correlation coefficient was 0.71 after removing 3 outlier residues (25, 91, 94) which they say were not compatible with their method.

We then applied REX to the MS dataset to infer residue level deuterium uptake. We first computed the correlation between our inferred uptake from REX and the experimentally observed NMR uptake, we filtered out the zero uptake data and residues which had zero redundancy to avoid inflating the correlation. We obtain a correlation of 0.72 despite not removing any outlier measurements and retaining contradicting measurements.

We further sought to assess REX's coverage. A well calibrated Bayesian analysis would imply that a 95% credible band contains 95% of the measurements. Hence, for each residue we computed how much of the data was covered by the credible bands, we found that in most cases it was 100% with a handful of residues falling below that (see figure 1)

Figure 1. REX's coverage, the percentage of HDX data points that fall within the credible bands, for each residue measured by NMR.

The average coverage was 91% which is slightly anti-conservative but respectable given the contradictions and limitations we highlighted earlier. Curiously, we found that the residues where we see poor coverage do not correspond to Stofella et al's outliers indeed the coverage for those residues was 25 (100%), 91 (75%) and 94 (100%) - figure 2 below. Instead we found residue 61 (which has a redundancy of 1) and residue 96 (which had a redundancy of 2) the poorest modelled. The peptides with contradicting uptakes with the NMR data covered residues [28-30] and [84-89]. The second contradicting peptide may explain the worse calibration for those residues and could indeed propagate errors to more distal residues (see further discussion below to another of the reviewers comments).

Figure 2. NMR data (black curves) with REX inferred confidence bands from MS data of deuterium uptake.

We then computed the average reconstruction error (ARE) between REX's estimates and the MS peptide level data to see if poor reconstruction could explain the differences between the analysis (see figure 3):

Figure 3. Average reconstruction error (ARE) for each residue between REX and the mass-spectrometry data.

We found that in general, aside from a few residues, modeling error was low. We conclude that benchmarking methods on this dataset might be misleading given the contradictions and data quality, as approaches that are highly faithful to the mass-spectrometry data will still contradict the NMR results. Hence, we have opted not to add this analysis to the main text but interested readers will still be able to access this information via the public peer review file. Adapting REX to include NMR constraints is left as future work.

- (1) Moulick R, Das R, Udgaonkar JB. Partially Unfolded Forms of the Prion Protein Populated under Misfolding-promoting Conditions: CHARACTERIZATION BY HYDROGEN EXCHANGE MASS SPECTROMETRY AND NMR. *J Biol Chem.* 2015 Oct 16;290(42):25227-40. doi: 10.1074/jbc.M115.677575. Epub 2015 Aug 25. PMID: 26306043; PMCID: PMC4646174.
- (2) Stofella, Michele, et al. "High-resolution hydrogen–deuterium protection factors from sparse mass spectrometry data validated by nuclear magnetic resonance measurements." *Journal of the American Society for Mass Spectrometry* 33.5 (2022): 813-822.

-Rex relies on overlapping peptides to infer residue level values. Please comment on the minimum redundancy needed to have confident results.

This is an important question raised by the reviewer, the level of redundancy is already incorporated into how our inferences are performed (using sums rather than averages) as explained in paragraph three of section 2.4. However, this does not fully answer the question of the minimal value of the redundancy needed to see confident results. In principle a redundancy of one is sufficient but will depend on other factors (such as peptide variability) as outlined in more detail to review in a reply to another of their questions.

Using our previous simulation study experiment (120 simulations) across a range of protection effects and scenarios, we examined how the redundancy of a residue chosen for perturbation impacted our results. As before, we calculated the absolute distance from the residue with high probability of change to the residue chosen for perturbation (lower is better) and also the probability of change of the residue chosen for perturbation (higher is better).

The analysis is displayed in the following figure:

We observe no obvious relationship between redundancy and quality of results, though data at the lowest redundancy is sparse. We tested the relationship between the calculated metrics and redundancy using a linear model. Using this analysis, we found no significant relationship between redundancy and these metrics. Though this is a surprising observation, we believe that it is not just redundancy but a number of factors - including peptide-level

variance which contributes to final inference quality. This is elaborated in a response to another of the reviewer's questions.

We thank the reviewer and have added the above figure to the supplementary material along with the following text:

To determine the relationship between redundancy (number of peptides that overlap a particular residue) and the quality of our results, we examined whether the distance from the perturbed residue to the residue with highest probability was affected by redundancy. Figure S5 shows that this quantity does not vary with redundancy. We also examined whether the probability associated with the perturbed residue varied with redundancy. Again we found no relationship between the two quantities. Based on our analysis, while redundancy of 1 can work, we recommend a minimum redundancy of 2-3 for reliable results in most experimental conditions.

-Please clarify what data is being simulated for Figure 5. What are the expected results there? Do the regions showing protection correlate with expected results?

We appreciate the reviewer request for clarity as it raises an important and subtle point. In the simulation, we perform only a few steps.

Step 1: Randomly select a residue

Step 2: Identify all peptide overlapping that residue

Step 3: Reduce deuterium percentage of those overlapping residues by specified percentage

Step 4: Apply REX

The following figure, we hope clarifies (included in the supplementary material):

Now we believe the reviewer is asking: why REX inferred protection effects in regions even if none were simulated?

This represents an important but underappreciated aspect of HDX analysis. If a peptide displays a protection effect (reduction in deuterium uptake) but overlapping residues display no change (zero reduction in deuterium), then since the uptake for a peptide is the sum of individual residues, if we are to maintain the zero sum for that overlapping peptide there must be a positive (increase in deuterium uptake) effect to compensate. This is despite no deprotection effect at the level of peptides, hence we believe there are many silent effects such as this being ignored in HDX experiments. If neighboring peptides also have no change this silent effect simply propagates unseen at the residue level. This is because the unseen deprotection effect must be then balanced by a corresponding protection effect. This is exactly the effect we observe in our analysis and we hope the following figure clarifies:

Hence, it is perhaps surprising to see these protection effects despite not simulating them but we believe they are valid. In our statistical analysis, to be conservative, we assumed these were false positive effects despite believing they are valid. We have added sections to the supplementary material to explain this phenomenon. Supplementary Fig. S4 demonstrates that we identify the correct regions with high probability.

-Residue resolved HDX, shows Rex being able to show differences when a 20% reduction in deuterium uptake is simulated. Can you comment on what is the minimum deuterium uptake reduction Rex needs to be able to identify the change? Most allosteric effects in proteins show minimal HDX change.

This is a valuable question from the reviewer. There are a number of effects that determine the REX's sensitivity to detect a change. The answer is complex because the minimum effect will depend on several factors that ultimately contribute to the inferred standard deviation in a complex manner. This will be dependent on the redundancy structure, the redundancy value, the peptide level variability, the lengths of the peptides covering the region and the number of replicates. Based on statistical principles, we can derive detection thresholds as shown in the following table. We use the fact that to measure at 95% confidence the difference must be greater than twice the standard deviation and at 99% confidence the difference must be greater than three times the standard deviation. The inferred value of Sigma measures the standard deviation and we quote results for typical lengths of peptides (5,10,15), with linear dependence on the length of the peptide. For example, with typical measurement precision (Sigma = 0.001) and moderate peptide length (10 residues), REX can detect changes as small as 2% deuterium uptake difference at 95% confidence.

Peptide Length	5	10	15	5	10	15
Confidence Level	95%	95%	95%	99%	99%	99%
Sigma						
0.001	0.01	0.02	0.03	0.015	0.03	0.045
0.002	0.02	0.04	0.06	0.03	0.06	0.09
0.003	0.03	0.06	0.09	0.045	0.09	0.135
0.005	0.05	0.1	0.15	0.075	0.15	0.225
0.01	0.1	0.2	0.3	0.15	0.3	0.45

We found that the level of Sigma (parameter in REX) is typically lower than 0.001 for most datasets and sometimes an order of magnitude smaller. We thank the reviewer for their question as we believe these reference parameters will be useful for those using Rex. We have added these values to the supplementary materials.

Minor comments:

Page 10, 3rd paragraph: Change Serium Amyloid Protein for Serum Amyloid Protein.

Thank you for spotting this error.

Page 10, 3rd paragraph: "being able to concertina"?

We have replaced "concertina" with "extend"

Page 10, 4th paragraph: 25 models of cytochrome C refers to S12 not S11.

Page 11, 1st paragraph: AUROC refers to S13 not S12.

Check all Supp Figures numbering.

We have recompiled the document to correct these incorrect figure cross-references.

Figure 4. Add meaning of AP,P strands both in text and in figure legend.

We appreciate the reviewer spotting that these are undefined. In both the legend and text we have added that P corresponds to parallel and AP to antiparallel.

If possible, change the colouring of figure8, it's hard to identify dots.

We agree with the reviewer. We've used the same colors but improved the contrast of the colors and added text labels to reduce ambiguity amongst similar colors.

Methods section: Analytical column used Waters 130Å

Thank you for spotting the missing angstrom symbol due to a latex error.

Methods section: Please include detailed ionisation and IM settings used in the cIMS instrument

We have added the following details:

The instrument was operated in positive mode and resolution mode (V-mode), with a capillary voltage of 3.0 kV, and sample cone voltage of 20 V. The cyclic ion mobility separation was completed in a single pass using an ADC start delay of 13 ms and two pushes per bin, and a TW static height of 22 V, with sequence phases of 10 ms inject, 3 ms separate, and 34 ms eject/acquire.

HDMSE mode was enabled for data-independent acquisition for the peptide mapping experiments. The spectra were collected between 50 and 2000 m/z, with transfer collision energy ramp from 18 V to 50 V. For labelling experiments, HDMS mode was used, with spectra acquired between 400 and 2000 m/z.

Reviewer #2 (Remarks to the Author):

Communications Chemistry
Manuscript ID: COMMSCHEM-24-0251-T

Title: Inferring residue level hydrogen deuterium exchange with ReX

Authors: Oliver M. Crook, Nathan Gittens, Chun-wa Chung, and Charlotte M. Deane

In this manuscript, the authors describe a new method for inferring residue level uptake patterns by leveraging peptide overlap, temporal data, and sequence

correlation. ReX treats HDX-MS as a multiple change-point problem within a Bayesian non-parametric framework, allowing for parameter inference, differential HDX confidence assessments, and uncertainty estimation. The paper introduces a novel computational approach that addresses significant challenges in interpreting HDX-MS data. The authors use the method to analyze the differential flexibility of BRD4's Bromodomains and to quantify conformational variations in LXRA induced by various small molecules.

The ability to infer residue-level HDX patterns with higher resolution and confidence is a valuable advancement. This will interest researchers in structural biology, particularly those studying protein dynamics and interactions. Overall, the paper is well-written and provides sufficient detail for reproducibility.

We appreciate the very positive review of our manuscript.

ReX's ability to adapt to data quality and quantify uncertainty sets it apart from existing methods. I recommend the paper for publication, subject to the authors addressing the following point:

We value the reviewer highlighting REX's key selling points and are glad that these have come across. We address the reviewers remaining points below:

1) Information on the computational efficiency and scalability of ReX, particularly for large protein datasets would be valuable. Specifically, could the authors comment on the trade-off between the accuracy of change-point detection and the computational speed of the algorithm? Highly accurate methods may be slower, while faster methods may sacrifice some accuracy.

We agree with the reviewers suggestion around the trade-off between computational time and accuracy. We believe the reviewer is asking about the specific run-time of the algorithms we compared, which we state below. We believe it more important to consider the number of observations rather than the size of the protein as this better reflects the amount of time the algorithm will take. Our BRD4 example is one the largest datasets we could find in the public domain and that contained over 2000 observations which took 4h to analyze whilst other methods are much faster. REX's computational time reflects its comprehensive Bayesian analysis providing full uncertainty quantification - the only method to provide statistical confidence intervals. The timing comparison was performed on an Intel Xeon Platinum 8268 CPU @ 2.90GHz. While REX requires more computation time, extracting equivalent confidence estimates from other methods would require bootstrapping or similar approaches, potentially increasing their computation time by orders of magnitude. The timing in the table should therefore be contextualized with the

information content of the results. Though we do appreciate that speed will reflect in the uptake of the methodology and plan to rewrite parts of the algorithm in C++ as part of future work.

Dataset	Observations	Length	Rex	PyHDX	P-Inv
Cytochrome-C	1156	105	1h 58	0h 05	< 1s
BRD4	2034	477	4h 05	0h 15	< 1s

Reviewer #3 (Remarks to the Author):

Summary

The paper by Crook *et al.* introduces and benchmarks ReX, an approach to analyze hydrogen-deuterium exchange (HDX) data using a Bayesian non-parametric model. The paper describes the probabilistic model and a Markov chain Monte Carlo (MCMC) algorithm to sample all model parameters. The probabilistic model generates peptide-level data from a representation of the polypeptide chain at a residue level. The residue-level parameters are combined via a latent change-point model. The number of change points is unknown and inferred from the data using a trans-dimensional MCMC algorithm. ReX is benchmarked on eight protein datasets and compared to other methods. Various analyses illustrate the strengths of the ReX approach.

Assessment

The paper by Crook *et al.* is innovative and statistically sound. The authors show convincingly that their approach is conceptually cleaner and performs better than existing approaches to analyze HDX-MS data. Therefore, in principle, the article should be considered for publication. However, there are various issues with the current version that should be addressed by the authors.

We are extremely pleased to hear these positive comments from the reviewer and appreciate their time in highlighting various places in which the manuscript can be improved. A point-by-point response to the reviewer is provided below.

Major issues

Probabilistic model and inference method

The description of the model (section 4) should be improved.

1. Being rather unfamiliar with HDX-MS data, it took me some time to understand the role of time t . The exposition of the model would be clearer, if you introduced exposure times t_m such that the data are $y_i(t_m)$ and i is an index enumerating the peptides, m an index enumerating different exposure times. Moreover, it would be useful to make the time dependence of μ_r explicit in Eq. (1).

We appreciate the spatio-temporal nature of data is complex and the text would benefit from additional clarity. Regarding $y_i(t_m)$ we have edited the text as follows:

We let $y_i(t_m)$ be the observed deuterium incorporation measurements where i indexes peptides from $i = 1, \dots, N$ and $m = 1, \dots, M$ is an index enumerating the M exposure times t_1, \dots, t_M to heavy water.

Regarding μ we have made explicit the dependence is on time by introducing a t variable.

2. What is the physical justification for tying the residue-level parameters together via the change-point model? Does it mean that local segments have similar structural properties that affect hydrogen-deuterium exchange such as secondary structure and/or solvent accessibility? Shouldn't the τ_k be located at residues where these properties change significantly? Moreover, if the change points are roughly positioned where the local structure changes, shouldn't the number of change points K increase with the number of residues R (which contradicts your choice $\lambda/(R-1)$ for the Poisson rate resulting in a Poisson-distributed number of segments K independent of R)?

The reviewer raises a number of important clarifying points.

The justification for the change-point model is twofold:

The first is for statistical convenience to capture the non-exchangeability of residues (else we could use a mixture model) and avoid overfitting by putting a prior on the complexity of the model. The second is that there will be a number of physical reasons for the change points including those mentioned by the reviewer but also others given the complexity of hydrogen-deuterium exchange biophysics, including the hydrogen-bonding network. If a 3D structure for the protein of interest is not known then these properties won't typically be known. Furthermore, some change points will be as a result of where we happen to measure the peptides through proteolytic digestion. Whilst we find some correlation between these

various physical characteristics and the distribution of the change-points (figure 4), we found our model more accurate when we allowed the change-points to be inferred rather than pre-allocate them.

Regarding the reviewer's other point, we appreciate that the choice of λ is unclear. In our model, we use a Poisson process to model change point locations along the protein sequence. The rate parameter of this Poisson process is $\lambda/(R-1)$, where R is the number of residues. This parameterization means that λ represents the prior expected number of change points for the protein. We find it more convenient to reason in terms of the interpretable quantity, the number of change points, rather than the abstract rate parameter, hence our choice of notation. This makes it easy for users to specify their prior beliefs about how many change points to expect, rather than having to think about rates per unit length

The reviewer correctly notes that keeping λ fixed while increasing R would result in the same expected number of change points for proteins of different lengths, which may not be biologically realistic. In practice, we expect larger proteins to have more structural domains and transitions, suggesting λ should increase with protein size. We have added guidance that λ should typically be scaled with protein length to reflect this biological expectation."

We parameterise our model so that λ represents the prior expected number of change points rather than a rate parameter, which is more intuitive for practitioners. For larger proteins, λ should be increased proportionally to reflect the expectation that longer sequences contain more structural transitions.

3. Symbols d_r , q_r are introduced in Eq. (1) and in the following explanation, but are not explained properly (perhaps they are the same?).

Thank you for pointing out the confusion here. Indeed, there is no q_r in the model; it is d_r - a remnant of a previous manuscript iteration. We have also added the following text to ensure all the kinetic parameters are described clearly:

The rate parameter of the Weibull model is b_r and the stretch parameter is p_r , whilst the rate parameter of the exponential model is defined as d_r .

4. What are start_i , end_i introduced in Eq. (2)? I guess these are the start and end of the i -th peptide, but the paper does not explicitly explain this.

We did not correctly introduce $\text{start}_i, \text{end}_i$ and the reviewer is correct so we have included the following text:

start_i and end_i denote the start and end of i^{th} -peptide

5. I find your notation for the change-point model difficult to comprehend. One reason is that you use the symbol θ for both residue and segment-specific parameters. Why not introduce a label $\ell_r \in \{1, \dots, K\}$ for each residue r indicating to which of the K segments residue r belongs, or equivalently binary variables $Z_{rk} \in \{0, 1\}, \sum_k Z_{rk} = 1$ similar to the allocation parameters of a mixture model that would replace the indicator function $\mathbb{I}(\tau_{k-1} \leq r < \tau_k)$ in Eq. (3)? The residue-level parameters would then be obtained from the segment-specific parameters θ_k as θ_{ℓ_r} or $\sum_k Z_{rk} \theta_k$ replacing $\theta(r)$, and the likelihood could be formulated in terms of π_{ℓ_r}, b_{ℓ_r} , etc.

We agree with the reviewer that our notation was confusing and we have taken on board the reviewers suggested notation. One clarifying point we wish to make is that the notation using Z perhaps does not make it clear that the residues are not exchangeable and so neither are the parameters, hence the change-point model rather than a mixture model. Whilst we could clarify this, borrowing notation from mixture models may encourage this confusion. Furthermore, the non-exchangeability is explicit when we define the prior model using functional notation. Hence, we have re-written the change-point model section in the hope to be more clear using the reviewer's suggestion to introduce a label, noting that we use the notation s rather than ℓ to avoid confusion with the length-bias term l .

We have used the following text which we hope is clearer:

There are $K + 1$ intervals to consider between each change point and the endpoints.

First, let us introduce $s_r \in \{1, \dots, K+1\}$ indicating to which interval the residue r belongs with $s_i \leq s_j$ if $i < j$. Then if $s_r = k$ then for residue r the following inequality holds: $\tau_{k-1} \leq r < \tau_k$. Then we define $\theta(r) = \theta_{s_r}$, which takes on one of $K+1$ values in the set $\{\theta_1, \dots, \theta_{K+1}\}$ and so it is convenient to write the following prior function:

$$\begin{equation} \theta(r) := \theta_{\ell_r} = \sum_{k=2}^{K+1} \theta_k \mathbb{I}(\tau_{k-1} \leq r < \tau_k), \end{equation}$$

from which it is obvious to see that the prior model is a piece-wise constant process. The prior model specification is complete once we specify priors for each of the individual parameters. It is convenient to index the parameters using k to specify this prior structure and so we write $\theta_k = (\pi_k, p_k, b_k, d_k)$ to be the vector of parameters for residues r such that $s_r = k$. We specify the following hierarchical structure ...

6. Distributions \mathcal{G} (Gamma), \mathcal{B} (Beta), etc. in Eq. (4) are not introduced.

We have now included definitions as follows:

Where \mathcal{G} denotes the Gamma-distribution with the shape and rate parameterization and \mathcal{B} denotes the Beta distribution with two shape parameters.

7. Weibull kinetics: The standard Weibull distribution allows for shape parameters p_r that are larger than 1. Why do you restrict p_r to $[0, 1]$ by the choice of a Beta prior for p_r ? Moreover, it would have been helpful (at least to me) to understand the model better, if you explained that your model of $\mu_r(t)$ combines two cumulative distribution functions (of the Weibull and the exponential distribution) and that, by construction, $\mu_r(t) \in [0, 1]$.

This is a good interpretation from the reviewer and we have included that description with the following text.

The form of $\mu_r(t)$ is the combination of the cumulative distributions of the Weibull and exponential distributions and hence, by definition, is constrained to the interval $[0, 1]$. Here, a value of 0 for μ_r means that for no protein has the amide hydrogen for residue r exchanged with deuterium whilst 1 represents complete exchange.

Regarding the Beta prior on p_r which results in constraints between $[0, 1]$ is based on a physical assumption. Values greater than 1 imply exchange rates faster than the intrinsic exchange rates of that residue would allow. Literature reference values for p_r appear to vary between 0 and 1, with 0.74 sometimes used as a common reference parameter (See for example Nyugen et al. Reference Parameters for Protein Hydrogen Exchange Rates JASMS)

8. What is the motivation for using a Laplace distribution in the likelihood?

The distribution was chosen for convenience rather than a physical motivation. We found that whilst in most cases the Gaussian distribution works well, it was too susceptible to outlier measurements. We found that Laplace distribution to be more robust to the outliers because it uses an absolute distance rather than squared distance.

We have added the following text:

using a Laplace error model (because of its robustness to outliers)

9. I would be interested in more details about the MCMC algorithm and its performance. For example, how sensitive are the results to the particular choice of hyperparameters (d_α , d_β , m , v , etc.) in Eq. (4)? What are typical computation times?

We added the following sensitivity analysis to the supplementary material. This analysis demonstrates that REX results are generally robust to hyperparameter choices, with clear relationships between prior specifications and model behavior.

We performed a prior sensitivity analysis to understand how prior choices influenced the results. Whilst we could not consider all possible combinations, we selected a handful of prior parameters and considered combinations of these prior options. We chose parameters that we thought would have a large effect on the final outcomes. The parameter choices were $\lambda \in \{1, 10, 100\}$, $m \in \{-5, -4, -3, -2\}$, $p_{\text{shape}_2} \in \{50, 500\}$, $b_\beta \in \{20, 200, 2000\}$. We considered all combinations of these options along with our defaults resulting in 72 different parameter combinations. In each scenario, we summarised the results by computing the number of change points, the value of σ , and the maximum values of ARE and TRE indicated error levels. To increase prior sensitivity we reduced our Cytochrome C to a single replicate and focused on the first 50 residues. The full results table can be found in the Supplementary Data.

We first examined how the number of change points affected prior choices. In Fig. \ref{figure::prior_sens-supp} A, we see that increasing λ , the parameter that controls the prior on the Poisson process, increases the number of change points as expected. We also see that increasing b_β results in a decrease to the number of changepoints. This is because increased values of b_β encourages higher rate of exchange forcing uptake values close to 1 for all timepoints.

We then examined how prior choices were reflected in the inferred value of σ . In Fig. \ref{figure::prior_sens-supp} B, we see that increasing λ , has no effect on the values of σ . However, there is a clear, as expected, positive relationship between the

m , the prior mean on $\log \sigma$, and the values of σ (on the log scale). Given that m is a prior directly on $\log \sigma$ this is the intended effect of the prior.

Ultimately, how these prior combinations affect modelling accuracy is more complex. In Fig. C, we plot the maximum values of the ARE (Max ARE) which measures modelling accuracy. We see that as the number of changepoints increases then Max ARE increases. This suggests that whilst a larger number of changepoints leads to more granular results, it is also potentially allowing the model to overfit. We find that lower errors are generally found for lower number of change points (smaller λ), middle values of b_{β} , and lower values of m . Hence, we recommend avoiding prior combinations that lead to extreme uptake values and a large number of changepoints.

Computation times are addressed in our response to Reviewer #2.

10. It can be tricky to set up reversible jump MCMC (RJMCMC). How do you ensure convergence of your MCMC sampler? How do you choose the initial parameters from which the Markov chain is started?

Random initialization would be extremely inefficient for this model. To initialize the model, we first use pseudo-inversion to estimate residue-level deuterium uptakes from the peptide-level data. Pseudo-inversion leads to non-physical values so we project data outside the expected range $[0, \phi]$ onto the closest boundary. We then fit our chosen non-linear model using least-squares to obtain initial parameter estimates which are subsequently smoothed using a L_1 trend filter. This provides initial parameters for our sampler. We have added the following text:

Algorithm initialisation

To initialise the algorithm, we need to propose starting parameters for the model. Whilst, in principle this can be done randomly, we take a pragmatic approach to start near a reasonable set of parameters. We first use generalised-inversion to estimate residue-level deuterium uptakes from the peptide-level data. Generalised-inversion leads to non-physical values so we project data outside the expected range of $[0, 1]$ onto the closest boundary. We then fit for each residue our functional model using least-squares to obtain initial parameter estimates which are subsequently smoothed using a L_1 trend filter \citep{Kim::2009}. This provides initial parameters for our sampler.

Convergence

We use the Gelman-Rubin diagnostic (3) on 4 parallel chains to assess convergence of the RJCMCMC algorithm. 5,000 steps were performed each with 5 Reversible-Jump moves per step (e.g. 25,000 Reversible-Jump moves in total). Chains were inspected manually for convergence, removing uncovered chains and removing the first 3,000 iterations for burn-in. We then confirmed that the Gelman-Rubin diagnostic was close to 1 on the log likelihood and Sigma parameters. We found BRD4 converged faster, running only 2,500 steps (12,500 Reversible-Jump Moves), using 1,500 steps for burn-in. Illustrative plots are shown below:

Convergence of multiple chains for different proteins

(3) Gelman, A and Rubin, DB (1992) Inference from iterative simulation using multiple sequences, *Statistical Science*, 7, 457-511.

11. Your description of RJMCMC on page 27/28 you seem to suddenly use lowercase letters k to indicate the number of change points. I find this confusing. Moreover, you seem to always change the dimension of the model, because you only mention moves $k \rightarrow k+1$ and $k \rightarrow k-1$. Why are there no moves preserving the number of model parameters?

Indeed, we agree that's confusing - we have changed these to upper case. In section 4.6.4. (previously 4.7) we stated that dimension preserving moves are simply typical MCMC moves. We wished to only state the more detailed derivation for the dimension switching moves as they are non-standard. We acknowledge that the reader is waiting several pages for this point and so we have added the following text at the end of section 4.5 (page 27/28):

To update the parameters for fixed model dimension K , then one can use a standard MCMC move such as the Metropolis-Hastings algorithm.

Other analyses

* Validation of AlphaFold models: At the end of section 2.3 you suggest that ReX could be used to validate 3D structure predictions from AlphaFold. In particular, your confidence score seems to reliably pick out regions that have a high/poor model quality, which is nice. However, have you tried secondary structure prediction from sequence only to check if the HDX-MS data are providing information that goes beyond standard secondary structure prediction?

The reviewer raises an important question about an important control and we have now added this analysis (using the same random forest model to keep all else equal). We found that the HDX features give a modest improvement over the sequence only model. We have added the following text.

This was higher than a sequence-only baseline which achieved a median AUROC of 0.61 $([0.43, 0.78], 90\%$ confidence interval)

Minor issues

Typos / grammar lapses

* General: "change points" not followed by a noun (such as in "change-point model") should not be hyphenated

We have attempted to correct the numerous errors here.

* Page 10: "adding additional" sounds a bit redundant;

We have removed "additional"

* Page 12: "The greater the number of change points less correlated"

Thank you! Corrected to: "The greater the number of change points the less correlated the uptake is along the sequence dimensions"

* Page 13: replace "de-convolute" with "deconvolve" or simply "distinguish"?

Thank you, we have replace with distinguish

* Page 14: "to improve this results", "our analysis ... advocate"

These have now been corrected. "These results" "our analysis ... advocates"

* Page 21: "procrusties transform" should be "Procrustes transform", "the majority of residues contributions"

These issues have been fixed.

* Page 24: "this functional form is more elaborate that previous approaches"

This has been corrected. "Than previous approaches"

* Page 25: "The choice of rate ensure", "From standard analysis of Poisson process" ("the" missing).

These have been fixed. "Ensures", "The Poisson Process"

* Page 25: I find the following sentence a bit cryptic:

"We note that whilst the prior model is hence piece-wise constant the posterior distribution (the distribution of the parameters having observed the data) is not necessarily in the same model family."

I guess you are saying the following: The model parameters and μ_r obtained by, for example, averaging over the posterior distribution are not necessarily piece-wise constant, because the location and number of change points changes with each MCMC sample.

The reviewer's explanation is indeed clearer. We have edited these sentences to:

We note that whilst the prior model is piecewise constant, when we obtain the model parameters and μ_r by, for example, averaging over the posterior distribution they are not necessarily piecewise constant, because the location and number of change points changes with each MCMC sample.

* Page 28: "Contrusion of RJMCMC", "Probability of new dimesnion"

These spelling errors have been fixed.

* Page 29: "we the parameters θ and σ "

Thank you for spotting this error.

* Page 31: "which residue are contributing"

We have fixed the pluralisation.

Figures

* Not all figures seem to be useful. For example, figure 2 simply illustrates the fact that the benchmark is based on predicting an entire digest that was held-out during model inference. The space could be used, for example, to show more details of the actual inference such as MCMC parameter traces, marginal distributions, etc.

We appreciate the reviewer's suggestion regarding MCMC diagnostics. However, Figure 2 serves an important pedagogical purpose for our target audience in structural biology and HDX-MS, where cross-validation approaches are significantly underused despite their importance for method validation. The train/test split concept, while fundamental in machine learning, requires explicit illustration for this community. We believe this conceptual illustration provides more value to our readers than technical MCMC diagnostics, which are typically relegated to supplementary materials even in statistical journals. MCMC convergence diagnostics are readily available in our software package (<https://ococrook.github.io/RexMS/articles/ReX.html>).

* Figure 1 uses the symbols ϑ_1 , ϑ_2 , etc. whereas the exposition in section 4 uses ϑ_k or $\vartheta(r)$.

Due to rendering limitations in our graphics software, both symbols appear identical despite representing the same mathematical object (produces ϑ even when typing θ). We tried again in powerpoint but the aesthetic is also different. We decided to keep the graphic and text as is, given that most readers will not appreciate the subtle difference and we hope statisticians will appreciate they are indeed to be the same.

* Figure S13: Panels in bottom row show model 17 multiple times, whereas the other models are only shown once.

This is by intention to keep the ROC curves simpler with many comparisons Model 17 is the only example where strand, sheets and loops were retained after cross-validation.

Other minor issues

* Two meanings of λ : LASSO parameter and rate of the Poisson process that governs the change-point model.

Thank you for spotting this. We believe readers may find this confusing and so have change λ for the LASSO parameter to λ_{LASSO} to differentiate, whilst using notation that would be expected.

* You do sampling. What's final prediction? A posterior mean over different change-point models?

Indeed! We appreciate that we mention this crucial detail quite late in the text and so have added another note earlier for clarity at the end of section 2.1:

The RJMCMC algorithm samples from the posterior distribution of this model which we summarise using the posterior mean.

* What are the PDB codes of the structures shown in Figure 6?

Thank you for spotting this omission - it is PDB: 3ZYU, which we have now added to the figure legend.

Reviewer #4 (Remarks to the Author):

I co-reviewed this manuscript with one of the reviewers who provided the listed reports. This is part of the Communications Chemistry initiative to facilitate training in peer review and to provide appropriate recognition for Early Career Researchers who co-review manuscripts.

We appreciate the collaborative review process and the valuable feedback provided through this co-review arrangement.

Reviewer #2 (Remarks to the Author):

Thank you for providing a clear and thorough response to my query. Your plans to re-implement performance-critical components in C++ sound promising and might be critical for the wider adoption of the method. At this stage, I have no further questions or concerns.

We thank the reviewer for their comments.

Reviewer #4 (Remarks to the Author):

The authors have addressed most of my concerns.

I believe it's important for readers to know that there is only one dataset containing both NMR and HDX-MS measurements with fully deuterated controls, and that RexMS produces contradictory results due to the quality of the experimental data. Please briefly address that in the main text and refer to the full analysis in the peer review/supporting information file.

With that clarification, I believe the paper is ready for publication.

We agree with the reviewer. A comment has been integrated into the benchmarking dataset and the analysis has been added to the supplementary information.